# Proteolytic maturation of $\alpha_2\delta$ represents a checkpoint for activation and neuronal trafficking of latent calcium channels

Ivan Kadurin*, Laurent Ferron[†], Simon W Rothwell[†], James O Meyer, Leon R Douglas[‡], Claudia S Bauer[§], Beatrice Lana, Wojciech Margas, Orpheas Alexopoulos, Manuela Nieto-Rostro, Wendy S Pratt, Annette C Dolphin*

Department of Neuroscience, Physiology and Pharmacology, University College London, London, United Kingdom

*For correspondence: i.kadurin@ucl.ac.uk (IK); a.dolphin@ucl.ac.uk (ACD)

[†]These authors contributed equally to this work

Present address: [‡]Southampton Cancer Research UK Centre, Faculty of Medicine, University of Southampton, Southampton, United Kingdom; [§]Sheffield Institute for Translational Neuroscience, University of Sheffield, Sheffield, United Kingdom

Competing interests: The authors declare that no competing interests exist.

**Abstract** The auxiliary $\alpha_2\delta$ subunits of voltage-gated calcium channels are extracellular membrane-associated proteins, which are post-translationally cleaved into disulfide-linked polypeptides $\alpha_2$ and $\delta$. We now show, using $\alpha_2\delta$ constructs containing artificial cleavage sites, that this processing is an essential step permitting voltage-dependent activation of plasma membrane N-type (Ca$_V$2.2) calcium channels. Indeed, uncleaved $\alpha 2\delta$ inhibits native calcium currents in mammalian neurons. By inducing acute cell-surface proteolytic cleavage of $\alpha_2\delta$, voltage-dependent activation of channels is promoted, independent from the trafficking role of $\alpha_2\delta$. Uncleaved $\alpha_2\delta$ does not support trafficking of Ca$_V$2.2 channel complexes into neuronal processes, and inhibits Ca$^{2+}$ entry into synaptic boutons, and we can reverse this by controlled intracellular proteolytic cleavage. We propose a model whereby uncleaved $\alpha_2\delta$ subunits maintain immature calcium channels in an inhibited state. Proteolytic processing of $\alpha_2\delta$ then permits voltage-dependent activation of the channels, acting as a checkpoint allowing trafficking only of mature calcium channel complexes into neuronal processes.

## Introduction

The $\alpha_2\delta$ subunits of voltage-gated calcium channels (Ca$_V$) have been identified to be key proteins in synaptic function and synaptogenesis (*Dickman et al., 2008*; *Kurshan et al., 2009*; *Hoppa et al., 2012*; *Eroglu et al., 2009*; *Saheki and Bargmann, 2009*). Therefore an understanding of their basic mechanism(s) of action is of paramount importance. Although Ca$_V\alpha$1 subunits form the pore and determine the main functional and pharmacological attributes of the channels (*Catterall, 2000*), the Ca$_V$1 and Ca$_V$2 channels are associated with auxiliary $\beta$ and $\alpha_2\delta$ subunits (*Flockerzi et al., 1986*; *Witcher et al., 1993*; *Takahashi et al., 1987*; *Liu et al., 1996*). The $\alpha_2\delta$ subunits increase Ca$_V$ currents by a mechanism that is less well understood than that of the $\beta$ subunits, which have a chaperone function (*Van Petegem et al., 2004*; *Buraei and Yang, 2010*; *Dolphin, 2012*; *Zamponi et al., 2015*). The topology of the $\alpha_2\delta$ proteins was initially determined for skeletal muscle $\alpha_2\delta$-1, but is likely to generalize for all $\alpha_2\delta$ subunits (*Brickley et al., 1995*; *Gurnett et al., 1996*, *1997*). A single gene encodes each $\alpha_2\delta$ protein, which undergoes several post-translational processing steps, including proteolytic cleavage into disulfide-linked $\alpha_2$ and $\delta$ (*Davies et al., 2010*; *Jay et al., 1991*; *Ellis et al., 1988*; *De Jongh et al., 1990*). Both $\alpha_2$ and $\delta$ have been shown previously to be important for the function of $\alpha_2\delta$-1 to increase Ca$_V$ currents and influence the biophysical properties of the currents (*Gurnett et al., 1996*, *1997*). The structure of the skeletal muscle Ca$_V$1.1 complex has recently been determined at 3.6 Å by cryo-electron microscopy (*Wu et al., 2016*). It shows the interaction of the $\alpha_2\delta$-1 with several extracellular linkers in Domains I-III of Ca$_V$1.1. In particular

the metal ion adhesion site (MIDAS) motif of the von Willebrand factor-A (VWA) domain interacts with the extracellular loop between transmembrane segments 1 and 2 in Domain I.

Regarding the mechanism of action of $\alpha_2\delta$ subunits, we have recently shown that $\alpha_2\delta$-1 increases the density of $Ca_V2.2$ channels inserted into the plasma membrane by about two-fold in undifferentiated neuro2A (N2A) cells (*Cassidy et al., 2014*); however the increase in currents directly attributable to $\alpha_2\delta$-1 is much greater than this (*Hoppa et al., 2012*). In the present study, we therefore examined whether there was an additional step responsible for promoting $Ca_V2.2$ calcium current function by $\alpha_2\delta$ subunits, in addition to their demonstrated effect on calcium channel trafficking (*Cassidy et al., 2014*). We have specifically tested the hypothesis that this step involves the proteolytic processing of $\alpha_2\delta$-1. Our results show that proteolytic maturation of $\alpha_2\delta$ subunits is not only a prerequisite for voltage-dependent activation of calcium channels, but is also an essential checkpoint for trafficking of these mature calcium channels into neuronal processes.

## Results

### Mutation of six amino acids flanking the cleavage site in $\alpha_2\delta$-1 prevents proteolytic processing of $\alpha_2\delta$-1 into $\alpha_2$ and $\delta$

We identified the proteolytic cleavage site in rat $\alpha_2\delta$-1 to be between A and V in the sequence LEA//VEM, by homology with the published data in rabbit (*Jay et al., 1991*; *De Jongh et al., 1990*) (*Figure 1a,b*). This sequence is strongly conserved in mammals (*Figure 1—figure supplement 1*). We initially manipulated the cleavage site in $\alpha_2\delta$-1 (*Figure 1b*), in order to determine the extent of mutation required across this site required to completely prevent proteolytic processing. Transient expression in cell lines of wild type (WT) $\alpha_2\delta$-1 resulted in only partial cleavage into $\alpha_2$ and $\delta$ subunits, clearly observed only following deglycosylation (*Figure 1c*, lanes 1 and 3), compared to the complete proteolytic cleavage observed in brain (*Figure 1c*, lanes 2 and 4). Replacing LEAVEM in $\alpha_2\delta$-1 with a hexavaline (V6) sequence (*Figure 1b*) completely prevented its proteolytic processing, as shown by the absence of cleaved $\alpha_2$-1 on reducing gels of whole cell lysate (WCL) (*Figure 1d*). More conservative mutations did not effectively prevent the cleavage (data not shown). However, we found that the lack of proteolytic cleavage did not affect the ability of $\alpha_2(V6)\delta$-1 to bind [3]H-gabapentin (*Figure 1e*). Furthermore, the $\alpha_2(V6)\delta$-1, like wild-type (WT) $\alpha_2\delta$-1, was found to be concentrated in detergent-resistant membranes (DRMs; *Figure 1—figure supplement 2*), and the $K_D$ for [3]H-gabapentin binding in DRMs was unchanged, being $76.2 \pm 7.8$ nM for WT $\alpha_2\delta$-1 and $85.0 \pm 13.7$ nM for $\alpha_2(V6)\delta$-1 (*Figure 1e*), indicating that the uncleaved pro-form of $\alpha_2\delta$-1 is likely to be correctly folded.

### Prevention of proteolytic processing does not impair trafficking of $\alpha_2\delta$-1 to the plasma membrane

We then decided to control the proteolytic processing of $\alpha_2\delta$-1 by inserting a Human Rhinovirus (HRV) 3C-protease site (*Cordingley et al., 1990*) into $\alpha_2\delta$-1, in place of the endogenous cleavage motif, to form $\alpha_2(3C)\delta$-1 (*Figure 1b*). This substitution completely prevented its proteolytic cleavage (*Figure 1f*). However, $\alpha_2(3C)\delta$-1 was still expressed on the cell surface (*Figure 1f–i*) and in DRMs (*Figure 1—figure supplement 3*), at similar levels to WT $\alpha_2\delta$-1.

We have previously shown that the amount of proteolytically-processed $\alpha_2\delta$-1 is increased on the cell surface compared to the WCL in transfected tsA-201 cells (*Kadurin et al., 2012*). However, the present results show that this processing of $\alpha_2\delta$-1 is not essential for its trafficking to the plasma membrane.

### Prevention of proteolytic processing of $\alpha_2\delta$-1 abolishes $Ca_V2.2$ current enhancement

It has previously been established that co-expression of WT $\alpha_2\delta$-1 results in an enhancement of $Ca_V2.2/\beta1b$ currents in heterologous systems (for review see [*Dolphin, 2012*]). We could confirm this result using $\beta1b$-GFP to ensure that all cells examined contain $\beta1b$; WT $\alpha_2\delta$-1 produced an 8.6-fold increase of $Ca_V2.2/\beta1b$ currents at +10 mV (*Figure 2a,b*). Strikingly, no increase was observed when $\alpha_2(3C)\delta$-1 was co-expressed with $Ca_V2.2/\beta1b$ (*Figure 2a,b*; *Figure 2—figure supplement 1*). This is in contrast to the ability of $\alpha_2(3C)\delta$-1 to reach the plasma membrane itself. In agreement with this,

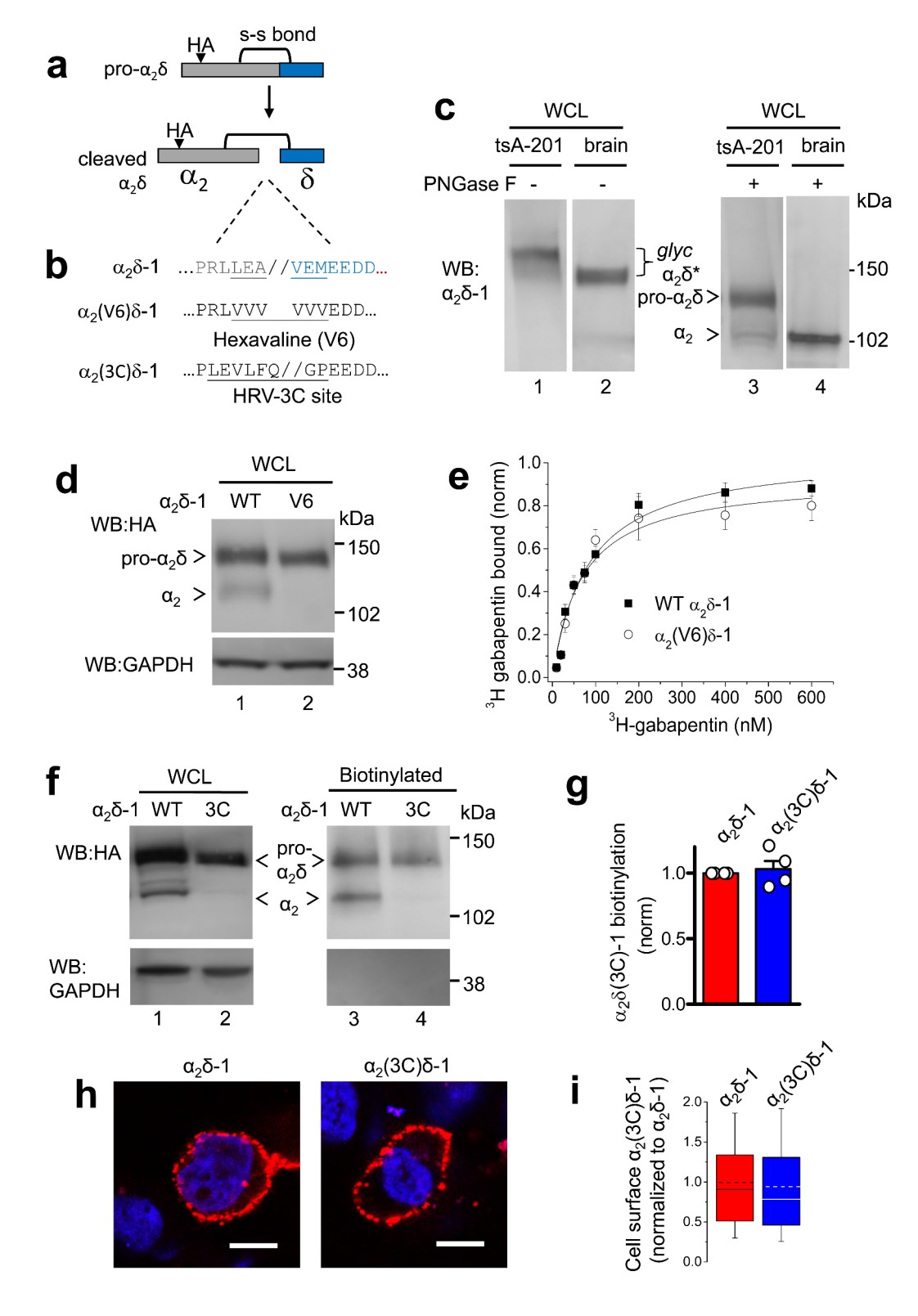

**Figure 1.** Effect of mutation of $\alpha_2\delta$-1 to disrupt the proteolytic cleavage site. (**a**) Cartoon of uncleaved pro-$\alpha_2\delta$-1 and cleaved $\alpha_2\delta$-1, showing the approximate position of inserted HA tag and disulfide bonds between $\alpha_2$ (grey) and $\delta$ (blue). (**b**) Rat $\alpha_2\delta$-1 sequence at the identified cleavage-site. The underlined sequence (including LEA//VEM in $\alpha_2\delta$-1) was mutated to a V6 or 3C-protease motif. (**c**) Comparison of glycosylated (lanes 1, 2) and deglycosylated $\alpha_2\delta$-1 (lanes 3, 4), expressed in tsA-201 cells (lanes 1, 3) or present in the brain (lanes 2, 4), showing resolution between pro-$\alpha_2\delta$-1 and

*Figure 1 continued on next page*

*Figure 1 continued*

the cleaved form of $\alpha_2\delta$ after deglycosylation. $\alpha_2\delta^*$ indicates glycosylated species, and pro-$\alpha_2\delta$ and $\alpha_2$ indicate deglycosylated proteins. Uncleaved pro-$\alpha_2\delta$-1 is observed in transfected cells, but not brain. Proteins visualized with $\alpha_2$-1 Ab. (d) $\alpha_2\delta$-1-HA (left) and $\alpha_2$(V6)$\delta$-1-HA (right) expressed in tsA-201 cells; proteins deglycosylated with PNGase-F. Upper panel: HA-blots, lower panel: endogenous GAPDH loading control. (e) Normalized binding curves, using DRM fractions from transfected tsA-201 cells, for $^3$H-gabapentin binding to WT $\alpha_2\delta$-1 (■, n = 4) and $\alpha_2$(V6)$\delta$-1 (○, n = 4). Mean (± SEM) data are fit by hyperbolae with $K_D$ of 82.3 and 68.3 nM, respectively. (f) Imunoblot analysis of deglycosylated $\alpha_2\delta$-1-HA and $\alpha_2$(3C)$\delta$-1-HA. WCL input (lanes 1, 2) and cell-surface biotinylated material (lanes 3, 4) are shown. Upper panel: HA-blot, lower panel: endogenous GAPDH. (g) Mean ± SEM (and individual data points) of $\alpha_2$(3C)$\delta$-1 (blue) cell surface levels, measured as a proportion of biotinylated: total protein, normalized to control (WT $\alpha_2\delta$-1; red), for 4 experiments including that shown in (f). p=0.5057, one sample t test. (h) Cell-surface expression of $\alpha_2\delta$-1-HA (left) and $\alpha_2$(3C)$\delta$-1-HA (right) in non-permeabilized tsA-201 cells, using HA Ab (red). Nuclei visualized with DAPI (blue). Scale bars 20 μm. (i) Box and whisker plots for quantification of $\alpha_2\delta$-1 on cell-surface, from HA fluorescence, for experiments including (h), for WT $\alpha_2\delta$-1 (red) and $\alpha_2$(3C)$\delta$-1 (blue). N = 290 and 317 cells, respectively, from 3 separate transfections, normalized to WT $\alpha_2\delta$-1 in each experiment. p=0.259, Student's t test.

The following figure supplements are available for figure 1:

**Figure supplement 1.** Alignment of the proteolytic cleavage site in $\alpha_2\delta$-1, showing species conservation.

**Figure supplement 2.** $\alpha_2$(V6)$\delta$-1 is localised in DRMs to a similar extent to $\alpha_2\delta$-1.

**Figure supplement 3.** $\alpha_2$(3C)$\delta$-1 is localised in DRMs to a similar extent to $\alpha_2\delta$-1.

another uncleavable construct used, $\alpha_2$(V6)$\delta$-1, was also unable to increase $Ca_V2.2$ currents (*Figure 2—figure supplement 2*).

## Proteolytic processing of $\alpha_2\delta$-1 is not required for the interaction with $Ca_V2.2$, and trafficking of the complex to the plasma membrane in cell lines

We next examined whether $\alpha_2$(3C)$\delta$-1 is impaired in its interaction with $Ca_V2.2$, which would explain its inability to increase $Ca_V2.2$ currents. However, we found that $Ca_V2.2$ co-immunoprecipitates with $\alpha_2$(3C)$\delta$-1 to the same extent as WT $\alpha_2\delta$-1 (*Figure 2c*). Furthermore, using extracellularly-tagged $Ca_V2.2$ to quantify the channels inserted into the plasma membrane (*Cassidy et al., 2014*), we found that in undifferentiated N2A cells, the uncleaved $\alpha_2$(3C)$\delta$-1 remains capable of increasing the cell surface density of $Ca_V2.2$ channels (123% increase compared to $Ca_V2.2/\beta1b$ alone), by a similar extent to WT $\alpha_2\delta$-1 (140% increase; *Figure 2d,e*). $\alpha_2$(3C)$\delta$-1 also increased cell surface expression of $Ca_V2.2$ in the tsA-201 cells used for electrophysiology (*Figure 2—figure supplement 3*). We then asked whether the depolarized resting membrane potential of N2A cells might influence their ability to traffic $Ca_V2.2$. We therefore co-expressed the leak $K^+$ channel, TASK3 (*Kim et al., 2000*), to hyperpolarize their membrane potential (*Figure 2f*), in order to determine whether this would differentially affect $Ca_V2.2$ trafficking supported by WT $\alpha_2\delta$-1 and $\alpha_2$(3C)$\delta$-1. This was not the case, as a similar increase in $Ca_V2.2$ surface expression was observed in the presence of TASK3 for $Ca_V2.2$, with either WT $\alpha_2\delta$-1 or $\alpha_2$(3C)$\delta$-1 (*Figure 2g*).

Thus, when heterologously expressed in cell lines, $\alpha_2$(3C)$\delta$-1 cause an increase of $Ca_V2.2$ surface expression, but does not result in an increase of $Ca_V2.2$ currents (see cartoon in *Figure 2h*), strongly suggesting $\alpha_2$(3C)$\delta$-1 may play an inhibitory role on $Ca_V2.2$ function.

## Proteolytic processing of $\alpha_2\delta$-1 is essential for voltage-dependent activation of $Ca_V2.2$ Currents

In order to determine whether $\alpha_2$(3C)$\delta$-1 could be cleaved at the inserted 3C-protease site, we then co-expressed an ER-lumen-targeted 3C-protease together with $\alpha_2$(3C)$\delta$-1 (*Figure 3a*). This resulted in efficient proteolytic cleavage of $\alpha_2$(3C)$\delta$-1 into $\alpha_2$ and $\delta$, as shown by the appearance of a band of the size of the $\alpha_2$ moiety, which reached the cell surface (*Figure 3b*; *Figure 3—figure supplement 1a*). As expected, co-expression of an inactive form of 3C-protease with a mutation in the catalytic site (C147V) failed to affect $\alpha_2$(3C)$\delta$-1 (*Figure 3—figure supplement 1b*).

Next we addressed the important question of whether proteolytic cleavage by 3C-protease could restore the currents through $Ca_V2.2$ channels containing $\alpha_2$(3C)$\delta$-1 subunits. In order to do this, we

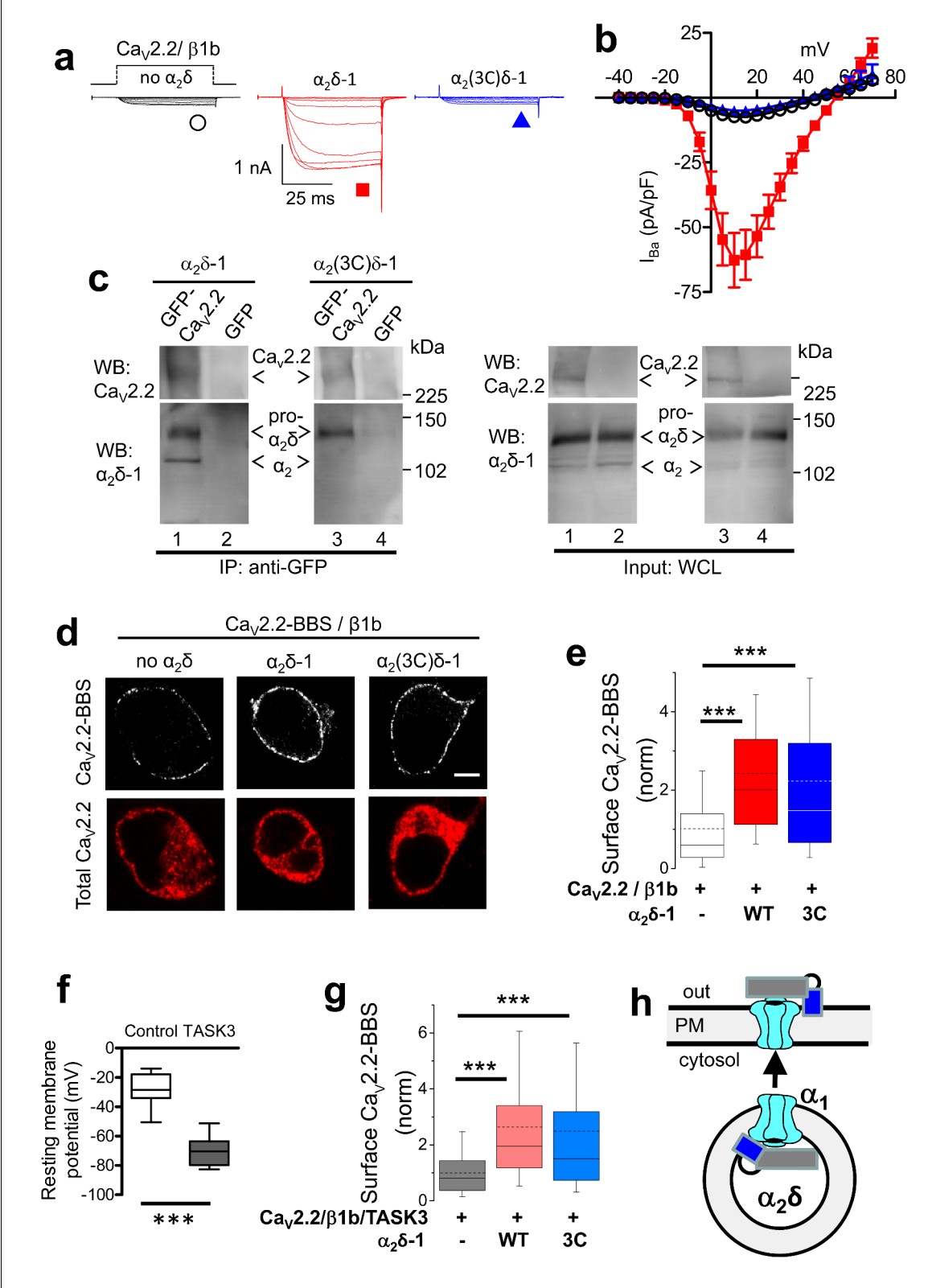

**Figure 2.** Effect of mutation of $\alpha_2\delta$-1 cleavage site to an HRV-3C site on cell-surface expression and functional properties of Ca$_V$2.2. (**a**) Example traces (−30 to +10 mV in 5 mV steps) for Ca$_V$2.2/β1b-GFP and no $\alpha_2\delta$ (black traces), WT $\alpha_2\delta$-1 (red traces) or $\alpha_2$(3C)$\delta$-1 (blue traces). Charge carrier 1 mM Ba$^{2+}$. Scale bars refer to all traces. (**b**) Mean (± SEM) *IV* curves for Ca$_V$2.2/β1b-GFP and no $\alpha_2\delta$ (black open circles, n = 14), WT $\alpha_2\delta$-1 (red squares, n = 34) or $\alpha_2$(3C)$\delta$-1 (blue triangles, n = 21). G$_{max}$: 0.25 ± 0.04, 1.91 ± 0.30 and 0.20 ± 0.03 nS/pF, respectively. V$_{50,act}$: 2.85 ± 0.68, 3.33 ± 0.46 and 3.89 ± 0.53

*Figure 2 continued on next page*

Figure 2 continued

mV, respectively. (c) tsA-201 cells transfected with GFP-Ca$_V$2.2 (lanes 1 and 3) or GFP (lanes 2 and 4), plus β1b, and either WT α$_2$δ-1 (lanes 1 and 2) or α$_2$(3C)δ-1 (lanes 3 and 4). Immunoprecipitation of GFP-Ca$_V$2.2 with anti-GFP Ab; WB with Ca$_V$2.2 II-III loop Ab (upper panels, lanes 1 and 3) produced co-immunoprecipitation (lower panels) of WT α$_2$δ-1 (lane 1) and α$_2$(3C)δ-1 (lane 3), revealed by α$_2$δ-1 mAb. Right panels: WCL input for lanes 1–4: upper panels, Ca$_V$2.2-GFP input; lower panels, α$_2$δ-1 input. All samples deglycosylated. (d) Immunocytochemical detection of cell-surface expression of Ca$_V$2.2-BBS, with β1b, and empty vector (left), WT α$_2$δ-1-HA (middle) or α$_2$(3C)δ-1-HA (right) in N2A cells. Upper panel: Ca$_V$2.2-BBS cell-surface staining prior to permeabilization (grey-scale); lower panel: total Ca$_V$2.2 after permeabilization (II-III loop Ab, red). Scale bar 5 μm. (e) Quantification of Ca$_V$2.2-BBS cell-surface expression (box and whisker plots) with empty vector (open bar, n = 206), WT α$_2$δ-1 (red bar, n = 191) or α$_2$(3C)δ-1 (blue bar, n = 181). Statistical differences: ANOVA and Bonferroni post-hoc test; ***$p<0.001$, compared to no α$_2$δ. (f) Resting membrane potential of control N2A cells (white bar, n = 16) and following expression of TASK3 (gray bar, n = 12). Box and whisker plots; ***$p<0.0001$, Student's t test. (g) Quantification of Ca$_V$2.2-BBS cell-surface expression in N2A cells co-expressing TASK3, with empty vector (gray bar, n = 70), WT α$_2$δ-1 (pink bar, n = 73) or α$_2$(3C)δ-1 (pale blue bar, n = 81). Box and whisker plots, statistical differences: ANOVA and Bonferroni post-hoc test, compared to no α$_2$δ; ***$p<0.001$. (h) Cartoon showing the ability of 'latent' Ca$_V$2.2 (cyan) plus α$_2$(3C)δ-1 (grey α$_2$, blue δ), to traffic to the plasma membrane (PM).

The following figure supplements are available for figure 2:

**Figure supplement 1.** Examination of effect of α$_2$(3C)δ-1 on Ca$_V$2.2/β1b calcium channel currents in tsA-201 cells.

**Figure supplement 2.** Examination of effect of α$_2$(V6)δ-1 on Ca$_V$2.2/β1b calcium channel currents in tsA-201 cells.

**Figure supplement 3.** Examination of effect of α$_2$(3C)δ-1 on Ca$_V$2.2/β1b cell surface expression in tsA-201 cells.

co-expressed 3C-protease with Ca$_V$2.2/β1b/α$_2$(3C)δ-1, and found that α$_2$δ-1-mediated Ca$_V$2.2 current enhancement was robustly rescued by the active protease, whereas the inactive mutant 3C-protease (C147V) had no effect (*Figure 3c–e*). This was not accompanied by any change in Ca$_V$2.2 trafficking (*Figure 3f,g*). The peak I$_{Ba}$ increase due to the 3C-protease was 5.4-fold at +10 mV, compared to inactive protease (*Figure 3c,d*). Co-expression of 3C-protease with α$_2$(3C)δ-1 also resulted in an increased activation rate of the Ca$_V$2.2 currents, to the same extent as WT α$_2$δ-1 (*Figure 3e*). As expected, when α$_2$(3C)δ-1 was not present, there was no increase in Ca$_V$2.2 currents attributable to 3C-protease (*Figure 3—figure supplement 2*).

In summary, expression of active, but not the inactive form of 3C-protease results in cleavage of α$_2$(3C)δ-1 and rescues enhancement of Ca$_V$2.2 currents by α$_2$(3C)δ-1, without any effect on trafficking, providing evidence that proteolytic processing of α$_2$δ-1 is required to promote voltage-dependent activation Ca$_V$2.2 channels.

## Replacement of the endogenous proteolytic site in WT α$_2$δ-3 with a 3C-protease site allows controlled processing of α$_2$δ-3 to rescue Ca$_V$2.2 currents

In order to further distinguish between the effects of α$_2$δ subunits on Ca$_V$2.2 channel trafficking and voltage-dependent activation, we also examined the behavior of α$_2$δ-3, because we surmised it might have a less prominent trafficking effect, as it contains an incomplete MIDAS motif (*Whittaker and Hynes, 2002*) in its VWA domain, which we have shown is essential for the trafficking and function of α$_2$δ-1 and α$_2$δ-2 (*Hoppa et al., 2012*; *Cassidy et al., 2014*; *Canti et al., 2005*). Indeed, we found that α$_2$δ-3 produced a much smaller increase than α$_2$δ-1 (31%, compared to ~140%) on Ca$_V$2.2 cell surface expression (*Figure 4a,b*), which we attribute to the absence of a complete MIDAS motif in α$_2$δ-3 (*Whittaker and Hynes, 2002*).

The site of proteolytic cleavage in α$_2$δ-3 has not been previously determined, although it is fully cleaved into α$_2$ and δ in vivo (*Davies et al., 2010*), and enhances Ca$_V$2.2 currents (*Davies et al., 2010*). We therefore first identified a potential cleavage site in α$_2$δ-3 by sequence alignment (*Figure 4c*), and then showed by mutational analysis that a V6 mutation across the predicted cleavage site prevented proteolytic cleavage, determined by the appearance of the δ-3 moiety in a reducing gel (*Figure 4—figure supplement 1*), and also prevented the functional effects of α$_2$δ-3 on Ca$_V$2.2 currents (data not shown). We then replaced this sequence in α$_2$δ-3 with the HRV-3C motif, to form α$_2$(3C)δ-3 (*Figure 4c*). We found that α$_2$(3C)δ-3 was expressed on the cell surface to a similar extent to WT α$_2$δ-3 (*Figure 4d*), despite the complete absence of proteolytic cleavage (*Figure 4d*,

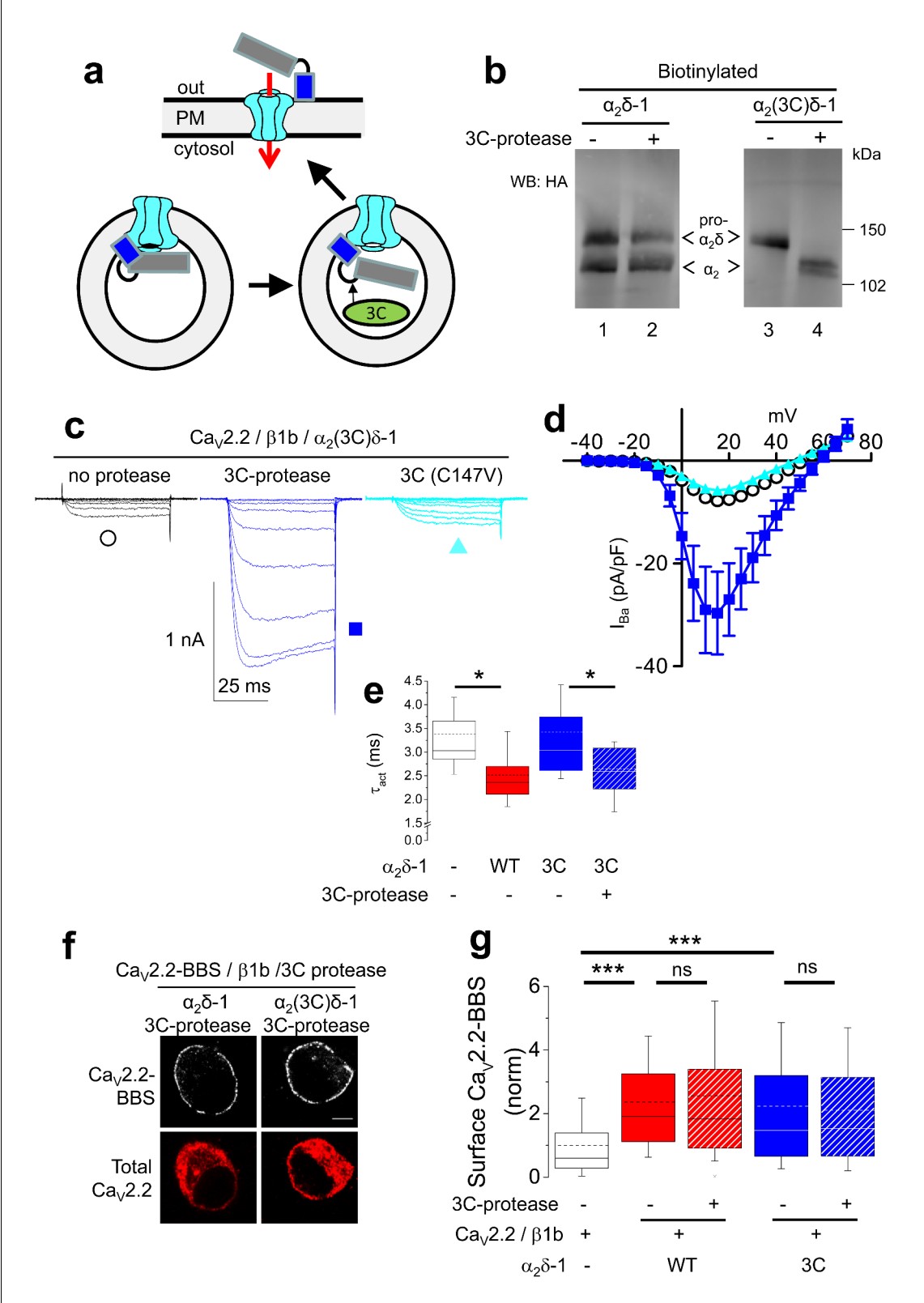

**Figure 3.** Effect of proteolytic cleavage of $\alpha_2\delta$-1 containing an HRV-3C cleavage site on cell-surface expression and functional properties of Ca$_V$2.2. (a) Cartoon showing intracellular cleavage by 3C-protease (green) of $\alpha_2$(3C)$\delta$-1 (gray/blue) associated with Ca$_V$2.2 (cyan). (b) Deglycosylated, cell-surface biotinylated fractions for $\alpha_2\delta$-1-HA (lanes 1 and 2) and $\alpha_2$(3C)$\delta$-1-HA (lanes 3 and 4), expressed in tsA-201 cells without (lanes 1 and 3) or with (lanes 2 and 4) 3C-protease. Representative of n = 4 experiments. WCL in *Figure 3—figure supplement 1* (c) Example traces (−30 to +10 mV steps) for Ca$_V$2.2/

*Figure 3 continued on next page*

*Figure 3 continued*

β1b-GFP/$\alpha_2$(3C)δ-1-HA and no protease (black traces), 3C-protease (blue traces) or inactive mutant 3C-(C147V) protease (cyan traces). Charge carrier 1 mM Ba$^{2+}$. Scale bars refer to all traces. (**d**) Mean (± SEM) *IV* curves for Ca$_V$2.2/β1b-GFP/$\alpha_2$(3C)δ-1-HA and no protease (black open circles, n = 26), 3C-protease (blue squares, n = 22) or 3C-(C147V)-protease (cyan triangles, n = 23). G$_{max}$: 0.26 ± 0.04, 0.90 ± 0.22 and 0.22 ± 0.03 nS/pF, respectively. G$_{max}$ values in the presence of the active 3C-protease were greater than in the absence of protease or presence of 3C-(C147V)-protease (Kruskal-Wallis test with Dunn's multiple comparison post-hoc test, p<0.05). V$_{50,act}$: 6.05 ± 0.82, 5.18 ± 0.74 and 6.20 ± 1.20 mV, respectively. (**e**) Time constant of activation ($\tau_{act}$) for I$_{Ba}$ at +10 mV for Ca$_V$2.2/β1b-GFP with no $\alpha_2$δ-1 (open bar, n = 14), WT $\alpha_2$δ-1-HA (red bar, n = 25), $\alpha_2$(3C)δ-1-HA (blue bar, n = 21) or $\alpha_2$(3C)δ-1-HA + 3C-protease (blue hatched bar, n = 17). Box and whisker plots. Statistical significance determined by ANOVA and Bonferroni's post-hoc test (*p<0.05). (**f**) Immunocytochemical detection of cell-surface Ca$_V$2.2-BBS, plus β1b, and $\alpha_2$δ-1-HA (left panel) or $\alpha_2$(3C)δ-1-HA (right panel), with 3C-protease. Upper panel: Ca$_V$2.2-BBS cell-surface staining (grey-scale), lower panel total Ca$_V$2.2 (II-III loop staining). Scale bar 5 μm. (**g**) Lack of effect of 3C-protease (hatched bars) on cell-surface expression of Ca$_V$2.2-BBS following expression of Ca$_V$2.2-BBS/β1b in N2A cells, with no $\alpha_2$δ (open bar, n = 206), WT $\alpha_2$δ-1 (red bar, n = 212), WT $\alpha_2$δ-1 and 3C-protease (red hatched bar, n = 192), $\alpha_2$(3C)δ-1 (blue bar, n = 181) or $\alpha_2$(3C)δ-1 and 3C-protease (blue hatched bar, n = 200). Box and whisker plots. Statistical differences determined by ANOVA and Bonferroni's post-hoc test (***p<0.001; ns: p>0.05).

The following figure supplements are available for figure 3:

**Figure supplement 1.** The effect of 3C-protease on $\alpha_2$δ-1 and $\alpha_2$(3C)δ-1 expressed in tsA-201 cells.

**Figure supplement 2.** Lack of effect of 3C-protease on Ca$_V$2.2/β1b currents expressed in tsA-201 cells.

lane 3). Importantly, when the 3C-protease was co-expressed, we found that $\alpha_2$(3C)δ-3 was still present on the cell surface, and was almost completely cleaved at the inserted HRV-3C site (*Figure 4d*, lane 4), although this had no effect on WT $\alpha_2$δ-3 (*Figure 4d,e*; *Figure 4—figure supplement 2a,b*). As we also found for $\alpha_2$(3C)δ-1, $\alpha_2$(3C)δ-3 did not increase Ca$_V$2.2 currents, whereas WT $\alpha_2$δ-3 produced a 6.6-fold increase (*Figure 4f*). However, inducing proteolytic cleavage of $\alpha_2$(3C)δ-3 by 3C-protease substantially rescued the enhancement of Ca$_V$2.2 currents, whereas the mutant protease 3C (C147V) did not (*Figure 4g,h*). The increase in peak I$_{Ba}$ due to the 3C protease was 2.7-fold at +10 mV, compared to inactive protease.

In summary WT $\alpha_2$δ-3 has a much smaller effect on Ca$_V$2.2 cell surface trafficking than WT $\alpha_2$δ-1, but proteolytic cleavage of $\alpha_2$(3C)δ-3 still produces a substantial increase in Ca$_V$2.2 currents.

## Proteolytic processing of $\alpha_2$δ subunits on the cell surface leads to rescue of Ca$_V$2.2 Currents

In order to further probe the role of proteolytic processing of the $\alpha_2$δ subunits on calcium channel currents, independent of their trafficking effects, we next turned to application of extracellular protease. However, we found that incubation of cells expressing $\alpha_2$(3C)δ-1 or $\alpha_2$(3C)δ-3 with purified 3C-protease did not result in their proteolytic cleavage at the plasma membrane (*Figure 4—figure supplement 3*, and data not shown). As an alternative approach, we inserted a thrombin cleavage site into $\alpha_2$δ-3 to produce $\alpha_2$(Th)δ-3 (*Figure 5a,b*). Importantly, we first showed that extracellular thrombin did not cleave WT $\alpha_2$δ-3 (*Figure 5—figure supplement 1*), and we found that $\alpha_2$(Th)δ-3 reached the cell surface as an uncleaved protein (*Figure 5c,d*; *Figure 5—figure supplement 1*). Furthermore, application of thrombin in the extracellular medium then produced marked cleavage of $\alpha_2$(Th)δ-3 on the cell surface (*Figure 5c*), and the optimal incubation period was about 60 min (*Figure 5—figure supplement 1*). This experiment was not attempted for $\alpha_2$δ-1 as preliminary studies revealed that $\alpha_2$δ-1 contained an ectopic thrombin cleavage site (data not shown).

In line with previous results, uncleaved $\alpha_2$(Th)δ-3 failed to mediate Ca$_V$2.2 current enhancement (*Figure 5e*). However, when thrombin was added to cells expressing Ca$_V$2.2/β1b and $\alpha_2$(Th)δ-3, prior to Ba$^{2+}$ current recording, it resulted in an acute increase in I$_{Ba}$ amplitude (*Figure 5e–g*), clearly demonstrating that cleavage of $\alpha_2$(Th)δ-3 has a direct effect to rescue Ca$_V$2.2/β1b/$\alpha_2$(Th)δ-3 currents. In contrast, extracellular application of thrombin had no effect on Ca$_V$2.2/β1b/WT $\alpha_2$δ-3 or Ca$_V$2.2/β1b currents (*Figure 5e–g*). Therefore, the functional role of $\alpha_2$δ-3 on Ca$_V$2.2 channels can be acutely restored at the plasma membrane via specific proteolytic cleavage of extracellular $\alpha_2$(Th)δ-3.

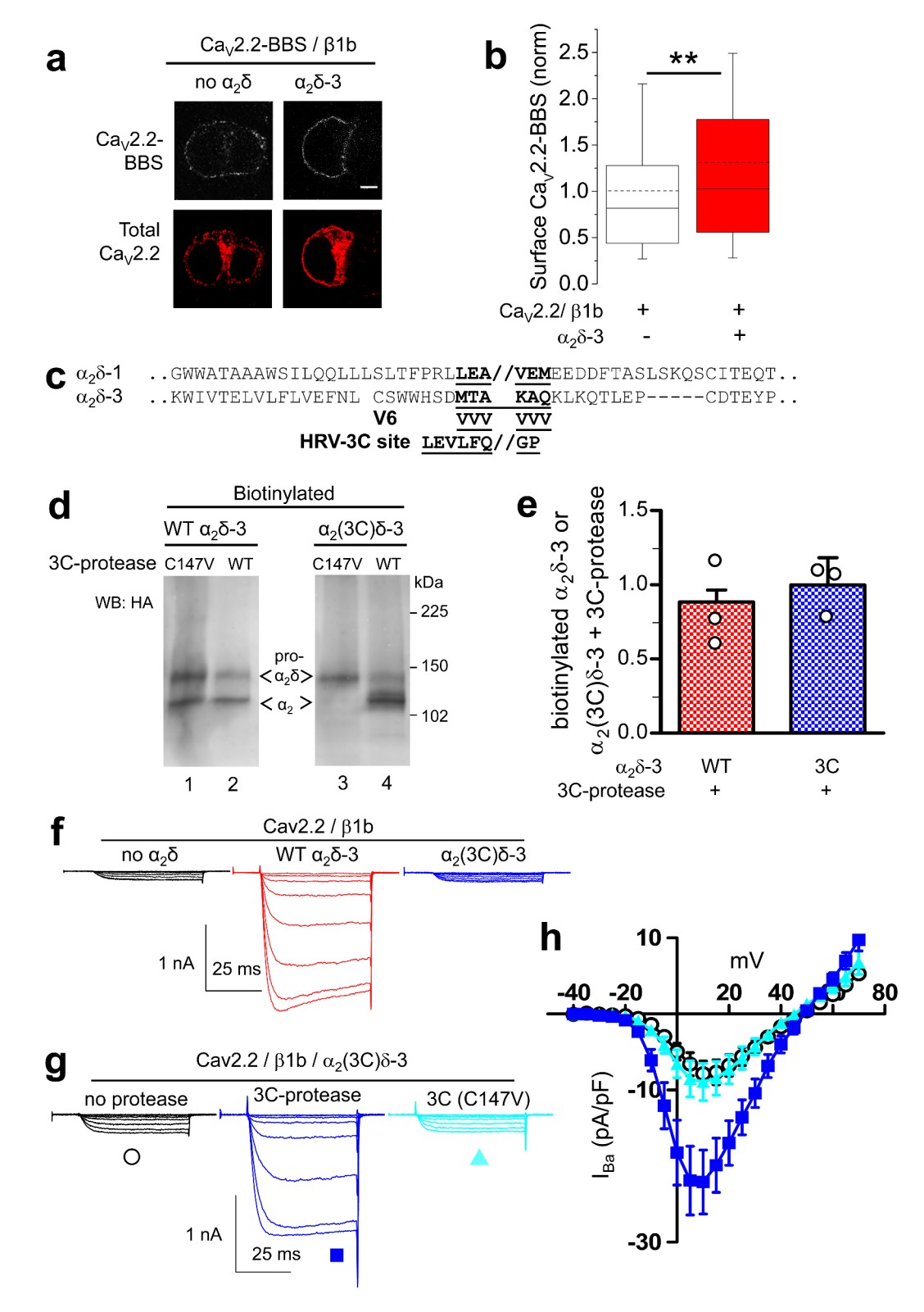

**Figure 4.** Effect of proteolytic cleavage of $\alpha_2\delta$-3 containing an HRV-3C cleavage site on cell-surface expression and functional properties of Ca$_V$2.2. (a) Images showing cell-surface Ca$_V$2.2-BBS (upper row, white), and total Ca$_V$2.2 (II-III loop Ab, lower row, red), for Ca$_V$2.2-BBS/$\beta$1b in N2A cells, with empty vector (panel 1) or $\alpha_2\delta$-3-HA (panel 2). Scale bar 5 $\mu$m. (b) Quantification (box and whisker plots) of effect $\alpha_2\delta$-3 on cell-surface Ca$_V$2.2-BBS following expression of Ca$_V$2.2-BBS/$\beta$1b with empty vector (open bar, n = 188) or WT $\alpha_2\delta$-3 (red bar, n = 188). Statistical difference determined by

*Figure 4 continued*

Student's t test, \*\*p=0.0028. (**c**) Alignment of $\alpha_2\delta$-3 sequence around the predicted cleavage site with $\alpha_2\delta$-1, showing weak homology. Underlined sequence (MTAKAQ) mutated to V6 or HRV-3C cleavage motif. (**d**) $\alpha_2\delta$-3-HA (lanes 1, 2) and $\alpha_2$(3C)$\delta$-3-HA (lanes 3, 4) expressed in tsA-201 cells, with either inactive (C147V, lanes 1, 3) or WT 3C-protease (WT, lanes 2, 4), cell-surface biotinylated and deglycosylated. Full WB and corresponding WCL in *Figure 4—figure supplement 3* (**e**) Quantification of cell-surface expression of WT $\alpha_2\delta$-3-HA (red speckled bar) and $\alpha_2$(3C)$\delta$-3-HA (blue speckled bar), with 3C-protease, normalized relative to inactive 3C-protease (C147V) for n = 3 experiments. Data are mean ($\pm$ SEM) and individual data points: p=0.4721 for WT $\alpha_2\delta$-3-HA and p=0.9513 for $\alpha_2$(3C)$\delta$-3 (1 sample t-test compared to respective control). (**f**) Example traces ($-30$ to +5 mV steps) for Ca$_V$2.2/$\beta$1b-GFP with no $\alpha_2\delta$ (black traces), WT $\alpha_2\delta$-3 (red traces) or $\alpha_2$(3C)$\delta$-3 (blue traces). G$_{max}$: 0.24 $\pm$ 0.03, 1.46 $\pm$ 0.22 and 0.21 $\pm$ 0.03 nS/pF respectively. V$_{50,act}$: 0.91 $\pm$ 1.015, 1.01 $\pm$ 0.85 and 4.03 $\pm$ 1.04 mV, respectively. (**g**) Example traces ($-30$ to +10 mV steps) for Ca$_V$2.2/$\beta$1b-GFP/$\alpha_2$(3C)$\delta$-3 and no protease (black traces), 3C-protease (blue traces) or inactive 3C-protease (C147V) (cyan traces). For (**f**) and (**g**), charge carrier 1 mM Ba$^{2+}$, scale bars refer to all traces. (**h**) Mean ($\pm$ SEM) *IV* curves for Ca$_V$2.2/$\beta$1b-GFP/$\alpha_2$(3C)$\delta$-3 without protease (black open circles, n = 28), with 3C-protease (blue squares, n = 29) or inactive 3C-(C147V)-protease (cyan triangles, n = 24). G$_{max}$: 0.28 $\pm$ 0.05, 0.70 $\pm$ 0.11 and 0.30 $\pm$ 0.08 nS/pF, respectively. G$_{max}$ for 3C-protease condition larger than other two conditions (Kruskal-Wallis test with Dunn's post-hoc test, p<0.05). V$_{50,\,act}$: 4.0 $\pm$ 0.7, 0.3 $\pm$ 0.7 and 1.5 $\pm$ 0.6 mV, respectively.

The following figure supplements are available for figure 4:

**Figure supplement 1.** Lack of cleavage of $\alpha_2$(V6)$\delta$-3.

**Figure supplement 2.** Effect of 3C-protease on expression and cleavage of $\alpha_2$(3C)$\delta$-3.

**Figure supplement 3.** Lack of effect of purified 3C-protease on expression and cleavage of $\alpha_2$(3C)$\delta$-3.

## Identification of the subcellular location of physiological cleavage of $\alpha_2\delta$-1 subunits in DRG neurons

The $\alpha_2\delta$-1 protein is extensively up-regulated in DRG neurons, following experimental peripheral nerve injury, such as spinal nerve ligation (SNL), and is trafficked to presynaptic terminals along the axon in intracellular trafficking vesicles (*Bauer et al., 2009*). This up-regulation allowed us to obtain sufficient protein to follow the processing of $\alpha_2\delta$-1. Following dissection of DRGs and associated axons, we observed that $\alpha_2\delta$-1 was fully glycosylated in the cell bodies of the DRGs, but remained only partially cleaved into $\alpha_2$ and $\delta$ (*Figure 6a*, tissue segment 3). We have shown previously by electron microscopy that within the cell body, $\alpha_2\delta$-1 is present both in the endoplasmic reticulum and at the plasma membrane, whereas in axons it is entirely associated with intracellular transport vesicles (*Bauer et al., 2009*). In the axons, we observed that proteolytic cleavage of $\alpha_2\delta$-1 was complete, both in the spinal nerve (*Figure 6a*, tissue segments 1, 2), and in the dorsal roots (*Figure 6a*, tissue segments 4, 5). This finding implies that all $\alpha_2\delta$-1 is proteolytically processed prior to intracellular trafficking into DRG axons. Indeed, our biochemical and cell fractionation data indicate that proteolytic cleavage occurs mainly during trafficking to the plasma membrane (*Kadurin et al., 2012*) (data not shown). However, this result opens the possibility that uncleaved $\alpha_2\delta$-1 may reach the cell surface in DRGs cell bodies and have a functional role there.

We then expressed WT $\alpha_2\delta$-1 in cultured DRG neurons in order to mimic its upregulation in the neuropathic state, and detected a significant amount of pro-$\alpha_2\delta$-1 in WCL (*Figure 6b*, lane 2). As expected, transfection of $\alpha_2$(3C)$\delta$-1 into DRG neurons resulted in abundant expression of the pro-$\alpha_2$(3C)$\delta$-1 form in WCL (*Figure 6b*, lane 3).

## Uncleaved pro-$\alpha_2\delta$-1 plays an inhibitory role in the calcium channel function

We then investigated whether uncleaved $\alpha_2\delta$-1 was inactive, or whether its presence might play an inhibitory role, as suggested by the fact that $\alpha_2$(3C)$\delta$-1 increases cell surface expression of Ca$_V$2.2 by about 2.5-fold in cell lines, but does not support any increase of Ca$_V$2.2 currents. We therefore examined the effect of $\alpha_2$(3C)$\delta$-1 expressed in cultured DRG neurons on native calcium channel currents (*Figure 6c*). We have previously shown that expression of WT $\alpha_2\delta$-1 in cultured DRG neurons caused an increase of native calcium currents (*D'Arco et al., 2015*) (shown with additional data in *Figure 6d*). Here, we found that expression of $\alpha_2$(3C)$\delta$-1 produced a marked reduction in

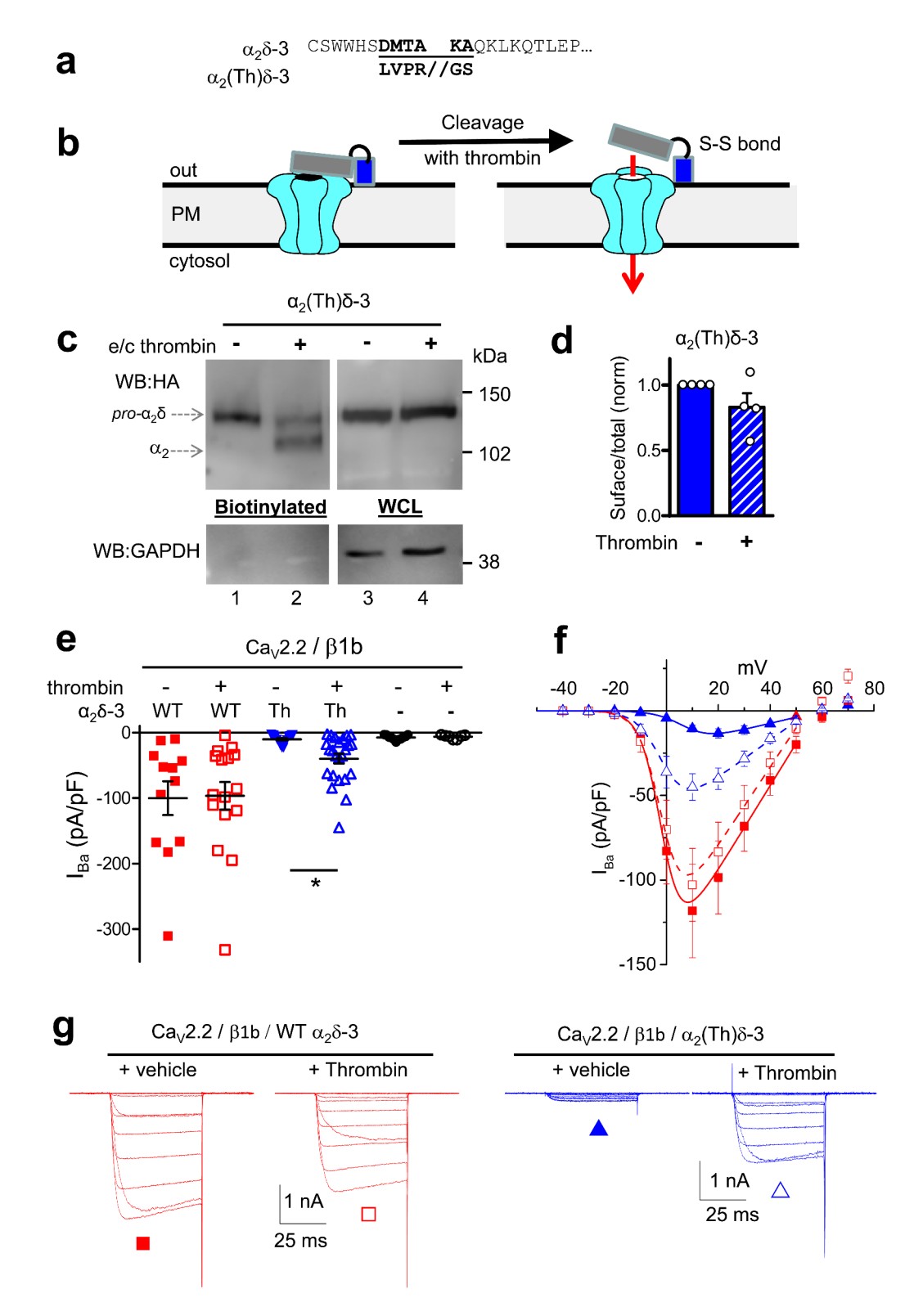

**Figure 5.** Effect of thrombin on the properties and function of $\alpha_2\delta$-3 containing a thrombin proteolytic cleavage site. (**a**) Sequence at $\alpha_2\delta$-3 cleavage site mutated to a thrombin cleavage site. (**b**) Cartoon of thrombin cleavage of $\alpha_2\delta$-3 on cell-surface. (**c**) Cell-surface biotinylation (left panel) shows efficient cleavage of cell-surface $\alpha_2$(Th)$\delta$-3-HA (lane 2), with no effect on total $\alpha_2$(Th)$\delta$-3-HA in WCL (right panel, lane 4). Samples were deglycosylated prior to loading. (**d**) Quantification of cell surface biotinylation experiments such as those shown in (**c**), indicating that thrombin does not decrease the

*Figure 5 continued on next page*

*Figure 5 continued*

amount of $\alpha_2(Th)\delta$-3-HA on the cell surface (hatched blue bar), normalized to vehicle application in each experiment (solid blue bar). Mean (± SEM) and individual data points for n = 4; p=0.2105, 1 sample t test. (**e**) Mean (± SEM) and individual data points of peak $I_{Ba}$ at +10 mV, for $Ca_V2.2/\beta1b$ with WT $\alpha_2\delta$-3 (red squares), $\alpha_2(Th)\delta$-3 (blue triangles) or no $\alpha_2\delta$ (black circles) and either no protease (closed symbols), or 60 min thrombin incubation (open symbols). The charge carrier was 2 mM $Ba^{2+}$. For data without or with thrombin, respectively, n = 12, 16 for WT $\alpha_2\delta$-3; 15, 24 for $\alpha_2(Th)\delta$-3 and 11, 7 without $\alpha_2\delta$, from at least 3 independent transfections. Statistical difference between thrombin and vehicle determined by Kruskal-Wallis ANOVA and Dunn's multiple comparison post-hoc test, *p<0.05. (**f**) Mean (± SEM) full *IV* curves for the same conditions as in (**e**) (excluding the no $\alpha_2\delta$ data), fitted with a modified Boltzmann equation to +50 mV. $G_{max}$ values (nS/pF) were 2.80 ± 0.61 (WT $\alpha_2\delta$-3; n = 10), 2.60 ± 0.55 (WT $\alpha_2\delta$-3 plus thrombin, n = 15), 0.43 ± 0.08 ($\alpha_2(Th)\delta$-3; n = 14), 1.26 ± 0.21 ($\alpha_2(Th)\delta$-3 plus thrombin, n = 21). $V_{50,act}$ values (mV) were +0.44 ± 1.71 (WT $\alpha_2\delta$-3), + 0.55 ± 1.28 (WT $\alpha_2\delta$-3 plus thrombin), +9.34 ± 0.08 ($\alpha_2(Th)\delta$-3), +0.87 ± 1.38 ($\alpha_2(Th)\delta$-3 plus thrombin). (**g**) Example $Ba^{2+}$ currents (from −50 to +60 mV) for the four conditions shown in (**f**).

The following figure supplement is available for figure 5:

**Figure supplement 1.** Controls for cleavage of $\alpha_2(Th)\delta$-3 by thrombin.

endogenous calcium channel currents in DRG neurons (by 43.2% at +10 mV; *Figure 6c,d*), supporting the hypothesis that pro-$\alpha_2\delta$-1 inhibits endogenous DRG calcium currents.

Furthermore, when $Ca_V2.2$-HA was transfected with $\beta1b$ and WT $\alpha_2\delta$-1 into DRG neurons, there was a clear increase in $Ca_V2.2$-HA on the cell surface of the DRG cell bodies after 2 days, compared to $Ca_V2.2$-HA/$\beta1b$ alone (*Figure 6e,f*). In contrast, for co-expression with $\alpha_2(3C)\delta$-1, the amount of $Ca_V2.2$-HA on the plasma membrane was the same as without exogenous $\alpha_2\delta$, rather than showing any inhibition (*Figure 6e,f*). This suggests that the inhibitory effect of $\alpha_2(3C)\delta$-1 on DRG calcium currents (*Figure 6c,d*) is not due to inhibition of channel trafficking to the somatic plasma membrane.

## Proteolytic cleavage of $\alpha_2\delta$-1 is required for trafficking of $Ca_V2.2$ Into neuronal processes

In view of our finding that all endogenous $\alpha_2\delta$-1 is proteolytically cleaved in axons but not cell bodies (*Figure 6a*), we then asked whether proteolytic processing was required for trafficking either of $\alpha_2\delta$-1 itself, or of associated $Ca_V2.2$, into neuronal processes. For this purpose, we used cultured hippocampal neurons which can be transfected after the establishment of extensive neurite outgrowth in culture. We found the surprising result that, when we co-transfected $Ca_V2.2/\beta1b$ with either WT $\alpha_2\delta$-1, $\alpha_2(3C)\delta$-1 or the corresponding empty vector, WT $\alpha_2\delta$-1 was essential for trafficking of $Ca_V2.2$ into the processes of hippocampal neurons, whereas in the presence of $\alpha_2(3C)\delta$-1, or without $\alpha_2\delta$, there was virtually no trafficking of $Ca_V2.2$ out of the soma (*Figure 7a,b*). Strikingly, the inability of $\alpha_2(3C)\delta$-1 to drive $Ca_V2.2$ into neuronal processes was reversed by co-expression of 3C-protease (*Figure 7c,d*).

The failure of $Ca_V2.2$ to traffic into the neurites in the presence of $\alpha_2(3C)\delta$-1 was not primarily due to a defect in the trafficking of the uncleaved $\alpha_2\delta$ subunit itself, since, when the $\alpha_2\delta$ subunits were transfected alone, WT $\alpha_2\delta$-1 and $\alpha_2(3C)\delta$-1 were both able to extensively access the hippocampal neurite compartment (*Figure 7e,f*). In contrast, when the localization of the $\alpha_2\delta$ subunits was examined following co-transfection with $Ca_V2.2/\beta1b$, the trafficking of $\alpha_2(3C)\delta$-1 into neurites was 37% lower than that of WT $\alpha_2\delta$-1 (*Figure 7g,h*), suggesting it has been retained by interaction with $Ca_V2.2$. Similar results to those with $\alpha_2\delta$-1 were obtained when comparing the ability of WT $\alpha_2\delta$-3 and $\alpha_2(3C)\delta$-3 to traffic $Ca_V2.2$ into neurites, which was also reversed by 3C-protease (data not shown). This further confirmed that the association of $Ca_V2.2$ with mature processed $\alpha_2\delta$ subunits is essential for trafficking of the complex into hippocampal neurites (*Figure 7—figure supplement 1*).

## Proteolytic cleavage of $\alpha_2(3C)\delta$-1 enhances $Ca^{2+}$ entry into hippocampal boutons

We then examined the effect of expression of $\alpha_2(3C)\delta$-1 alone in hippocampal neurons on action potential (AP)-induced $Ca^{2+}$ entry into presynaptic boutons. These were identified by co-expressed vesicle-associated membrane protein-mOrange2 (VAMP-mOr2), and $Ca^{2+}$ transients were measured using synaptophysin-GCaMP6f (sy-GCaMP6f, *Figure 8a*). The uncleaved $\alpha_2(3C)\delta$-1 reduced $Ca^{2+}$ entry in response to single AP stimulation in all experiments performed (*Figure 8b,c*), to 65.8 ± 9.1%

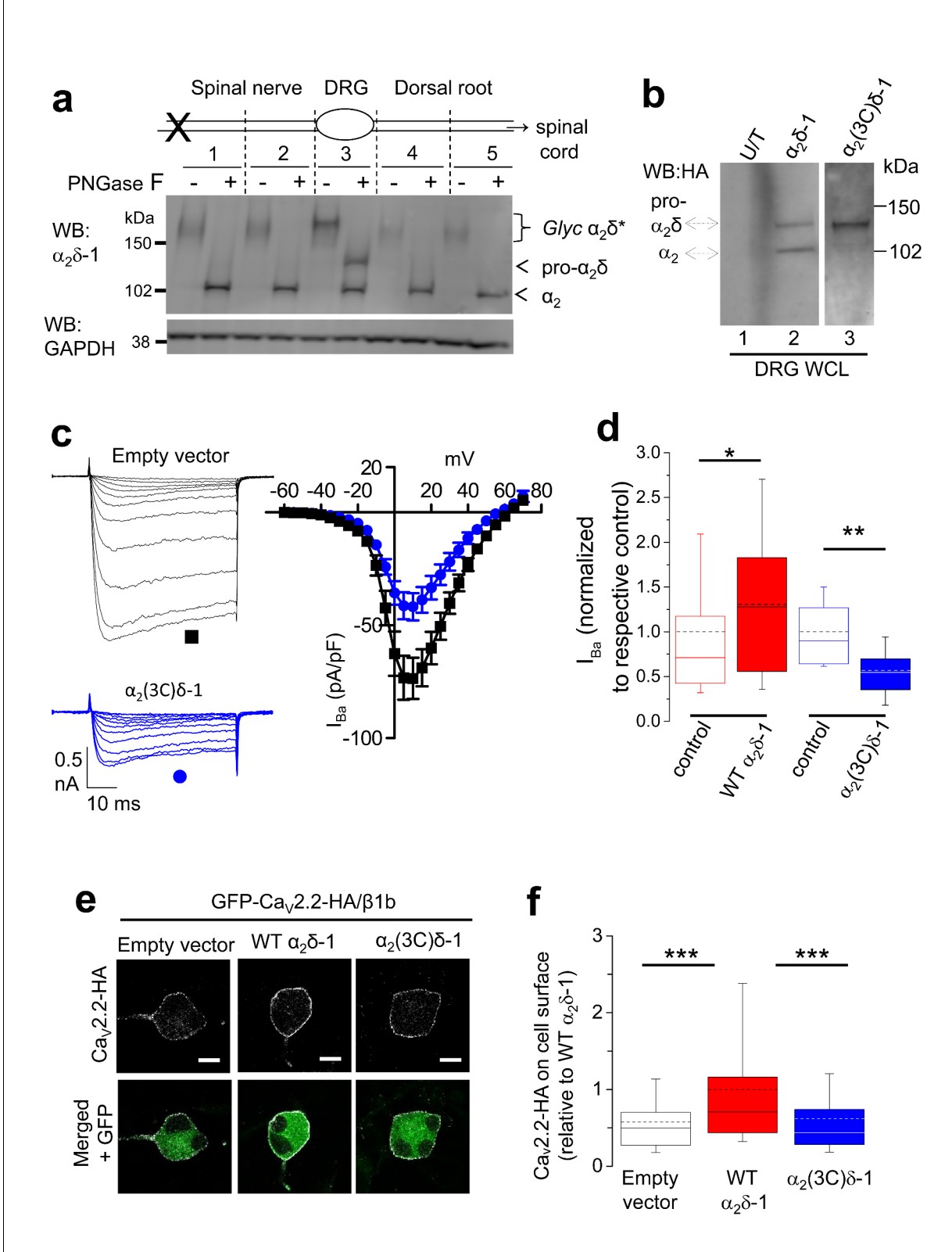

**Figure 6.** Proteolytic processing of endogenous $\alpha_2\delta$-1 and effect of exogenous expression of $\alpha_2\delta$-1 and $\alpha_2(3C)\delta$-1 in DRGs. (a) DRGs, spinal nerves and dorsal roots from rats, 4 days after SNL, were dissected and segmented according to the diagram. X marks site of ligation. Tissue was pooled from 4 rats. It was either treated or not with PNGase-F as indicated, and reduced with DTT; deglycosylation allows resolution of two $\alpha_2$-immunoreactive bands. Unprocessed $\alpha_2\delta$-1 is present only in the cell body compartment (segment 3) and is distinct from processed $\alpha_2$-1 (indicated by arrows). Lower blot is

*Figure 6 continued on next page*

*Figure 6 continued*

GAPDH loading control. (**b**) WCL for empty vector-transfected DRGs (U/T, lane 1); WT $\alpha_2\delta$-1-HA-transfected DRGs (lane 2); $\alpha_2$(3C)$\delta$-1-HA transfected DRGs (lane 3). WB: anti-HA. (**c**) Left: Example traces (−45 to +5 mV steps) for control (empty vector-transfected) DRG neurons (black traces, top) and DRGs transfected with $\alpha_2$(3C)$\delta$-1 (blue traces, bottom). The charge carrier is 1 mM Ba$^{2+}$. The scale bars refer to all traces. Right: Mean (± SEM) *IV* curves for control DRG neurons (black squares, n = 12) and DRGs transfected with $\alpha_2$(3C)$\delta$-1 (blue circles, n = 14), from 3 independent experiments. G$_{max}$ values were 2.20 ± 0.30 and 1.26 ± 0.14 nS/pF, respectively; p=0.0061 (Student's t test). V$_{50, act}$ values were −1.5 ± 1.2 and −1.5 ± 0.7 mV, respectively. (**d**) Comparison of normalized peak I$_{Ba}$ in control DRGs (open red bar, n = 55) and WT $\alpha_2\delta$-1-transfected DRGs (closed red bar, n = 54) including data from ***Figure 2d*** in, or comparison of control DRGs (open blue bar, n = 12) with $\alpha_2$(3C)$\delta$-1 transfected DRGs (closed blue bar, n = 14). Box and whisker plots. Statistical differences, Student's t test: *, p=0.048, **p=0.0067 compared to respective control. (**e**) Confocal optical sections (1 μm) showing GFP-Ca$_V$2.2-HA in non-permeabilized DRG neurons (top, white), when co-transfected with β1b and either empty vector (left), WT $\alpha_2\delta$-1 (middle) or $\alpha_2$(3C)$\delta$-1 (right). GFP fluorescence is shown in the merged lower panel. Scale bars: 10 μm. (**f**) Box and whisker plot of cell surface HA fluorescence density as a ratio of internal GFP density for Ca$_V$2.2-HA expression in DRG somata, transfected with empty vector (open bar, n = 81), WT $\alpha_2\delta$-1 (red bar, n = 133) or $\alpha_2$(3C)$\delta$-1 (blue bar, n = 159). ***p<0.001, 1 way ANOVA and Bonferroni post hoc test.

(n = 7, p=0.0093, 1 sample t test) of control. The inhibitory effect of $\alpha_2$(3C)$\delta$-1 on the response to a single AP was reversed by co-expression of 3C-protease in all experiments performed (***Figure 8d,e***), an increase of 81.5 ± 16.1% (n = 8, p<0.0001, 1 sample t test).

## Discussion

In this study, we have identified key roles for proteolytic processing of pro-$\alpha_2\delta$ subunits. Our main findings are: firstly that proteolytic cleavage of pro-$\alpha_2\delta$ represents an essential step for the expression of mature functional calcium channels. Here we show, using a combination of techniques, that the appearance of functional Ca$_V$2.2 channels associated with the proteolytic processing of $\alpha_2\delta$ subunits, occurs independently of changes in channel trafficking to the plasma membrane in undifferentiated cell lines. A key experiment is the demonstration that acute restoration of voltage-dependent activation by proteolytic cleavage of pro-$\alpha_2\delta$ can be induced at the cell surface by extracellularly applied protease. Secondly, we find that lack of proteolytic cleavage of $\alpha_2\delta$-1 represents a very significant barrier to trafficking of Ca$_V$2.2 channels into cultured hippocampal neuronal processes. Trafficking of the channel complex into neurites is dependent on an effect of mature cleaved $\alpha_2\delta$ on Ca$_V$2.2, rather than due primarily to the differential ability of mature $\alpha_2\delta$ to be trafficked and uncleaved pro-$\alpha_2\delta$ to be retained in the soma. This is clearly shown by the fact that uncleaved $\alpha_2$(3C)$\delta$-1 is fully able to traffic alone into neurites. Thirdly, we provide evidence for an inhibitory role for the pro-form of $\alpha_2\delta$-1. It is highly likely that proteolytic cleavage of $\alpha_2\delta$ could induce a conformational change, which would impact on its interaction with the $\alpha_1$subunit. The recent structure of the Ca$_V$1.1 complex resolves the domain structure of the $\alpha_2\delta$-1 subunit to contain the VWA domain and four tandem cache domains (***Wu et al., 2016***). The $\delta$ subunit contributes part of the fourth cache domain, and it is therefore possible that the domain organization would be modified by proteolytic cleavage of $\alpha_2\delta$.

In a previous study, we showed that WT $\alpha_2\delta$-1 increases the amount of Ca$_V$2.2 on the cell surface in N2A cells, by about two-fold (***Cassidy et al., 2014***). We have also found that heterologously-expressed $\alpha_2\delta$ subunits are only partially processed into $\alpha_2$ and $\delta$ in all expression systems examined, although proteolytic cleavage was much more marked at the plasma membrane (***Davies et al., 2010***; ***Kadurin et al., 2012***). The incomplete cleavage of heterologously expressed WT $\alpha_2\delta$ is in contrast to the complete processing of muscle and brain $\alpha_2\delta$-1 (***Jay et al., 1991***; ***Patel et al., 2013***) and cerebellar $\alpha_2\delta$-2 (***Davies et al., 2006***), and likely represents saturation of the processing enzyme(s). Thus, it was clear that determining the role of proteolytic cleavage of $\alpha_2\delta$ on calcium channel function would require additional strategies. A previous study found that various mutations around the cleavage site reduced, but did not abolish, calcium current enhancement by $\alpha_2\delta$-1 subunits (***Andrade et al., 2007***), leaving the role of proteolytic cleavage of $\alpha_2\delta$-1 an open question. Here we show conclusively that for both $\alpha_2\delta$-1 and $\alpha_2\delta$-3, mutations that prevent their cleavage into $\alpha_2$ and $\delta$ do not prevent the appearance of pro-$\alpha_2\delta$ on the cell surface. Furthermore, in the cell lines examined here, but not in neurons, Ca$_V$2.2 trafficking to the plasma membrane was enhanced by an uncleaved pro-form of $\alpha_2\delta$-1 to a similar extent as for WT $\alpha_2\delta$-1, utilising a mechanism that is independent of the plasma membrane potential. However, this led to the increased cell surface

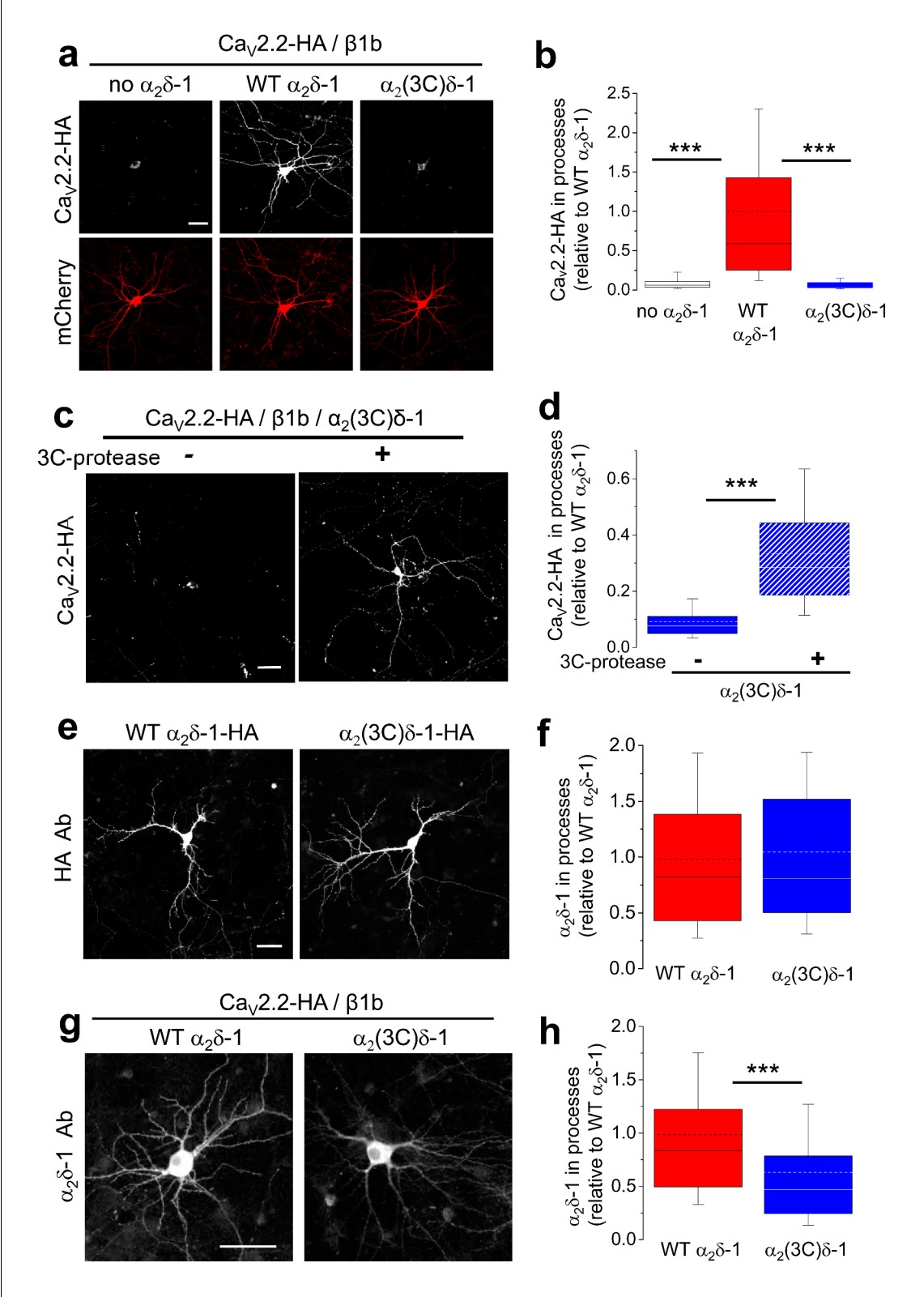

**Figure 7.** Effect of α₂δ-1 and proteolytic cleavage of α₂(3C)δ-1 on trafficking of Ca$_V$2.2 into hippocampal neurites. (a) Images showing Ca$_V$2.2-HA in permeabilized hippocampal neurons (top, white), when co-transfected with β1b, mCherry (bottom, red) and either no α₂δ (left), WT α₂δ-1 (middle) or α₂(3C)δ-1 (right). Scale bar: 50 µm applies to all images. (b) Box and whisker plots for Ca$_V$2.2-HA expression in processes without α₂δ (open bar, n = 136 processes from 29 cells), with α₂δ-1 (red bar, n = 147 processes from 27 cells) or with α₂(3C)δ-1 (blue bar, n = 109 processes from 22 cells). ***p<0.001, *Figure 7 continued on next page*

*Figure 7 continued*

1 way ANOVA and Bonferroni *post hoc* test. (c) Images showing $Ca_V2.2$-HA in permeabilized hippocampal neurons (white), when co-transfected with $\beta1b$, $\alpha_2(3C)\delta$-1 and mCherry (transfection marker, not shown), either without (left) or with (right) 3C-protease. Scale bar: 50 µm applies to both images. (d) Box and whisker plots for $Ca_V2.2$-HA expression in processes with $\alpha_2(3C)\delta$-1, transfected without (solid blue bar, n = 191 processes), or with 3C-protease (blue hatched bar, n = 187 processes). ***p<0.001, 1 way ANOVA and Bonferroni *post hoc* test. (e) Images showing WT $\alpha_2\delta$-1-HA (left) or $\alpha_2(3C)\delta$-1-HA (right) expressed in permeabilized hippocampal neurons (white), co-transfected only with mCherry (transfection marker, not shown). Antigen retrieval was used prior to the HA Ab. Scale bar: 50 µm applies to both images. (f) Box and whisker plots for expression in the processes of WT $\alpha_2\delta$-1-HA (red bar, n = 248 processes from 52 cells) and $\alpha_2(3C)\delta$-1 (blue bar, n = 263 processes from 51 cells). (g) Images showing $\alpha_2\delta$-1 in hippocampal neurons (white), when transfected with $Ca_V2.2$-HA, $\beta1b$, mCherry (transfection marker, not shown) and either WT $\alpha_2\delta$-1 (left) or $\alpha_2(3C)\delta$-1 (right). Antigen retrieval was used prior to the $\alpha_2\delta$-1 mAb. Scale bar: 50 µm applies to both images. (h) Box and whisker plots for $\alpha_2\delta$-1 expression in hippocampal processes, for WT $\alpha_2\delta$-1 (red bar, n = 221 processes) and $\alpha_2(3C)\delta$-1 (blue bar, n = 184 processes). ***p<0.0001, Student's t test.

The following figure supplement is available for figure 7:

**Figure supplement 1.** Cartoon showing processing and trafficking of $\alpha_2\delta$ in neurons.

expression of a calcium channel complex which appeared to be non-functional, since the uncleaved pro-$\alpha_2\delta$ did not enhance $Ca_V2.2$ calcium currents. Thus trafficking of $Ca_V2.2$ to the plasma membrane by $\alpha_2\delta$-1 subunits in cell lines can be uncoupled from the functional effects of $\alpha_2\delta$ subunits on voltage-dependent activation of the channels.

Previous in vitro studies have examined the effect of $\alpha_2\delta$ subunits on calcium channel currents resulting from several combinations of $Ca_V\alpha1$ and $\beta$ subunits. In whole-cell current recordings, $\alpha_2\delta$ subunits increase the maximum conductance from 3 to 10-fold, depending on subunit combination and conditions used (*Canti et al., 2005*; *Mori et al., 1991*; *Klugbauer et al., 1999*; *Hobom et al., 2000*; *Gao et al., 2000*; *Barclay et al., 2001*; *Hendrich et al., 2008*). However, they have no effect on single channel conductance, and for $Ca_V2$ channels there are only minor effects of $\alpha_2\delta$ subunits on most parameters relating to open probability (*Hoppa et al., 2012*; *Barclay et al., 2001*; *Brodbeck et al., 2002*; *Wakamori et al., 1999*; *Meir and Dolphin, 1998*), which would be unlikely to account for the large increase in whole-cell conductance. By contrast, a finding that is consistent with the increase in whole-cell current, is that the fraction of null traces was markedly reduced by $\alpha_2\delta$-1 in unitary current recordings from oocytes expressing $Ca_V2.2/\beta1b/\alpha_2\delta$-1 (9% null traces), a 3–4-fold decrease, compared with those expressing $Ca_V2.2$ alone (39% null traces) or $Ca_V2.2/\beta1b$ (28% null traces) (*Wakamori et al., 1999*). This observation suggests that $\alpha_2\delta$ shifts the equilibrium towards active modes of the channel, from an inactive null mode represented by the long closed state. Furthermore, it has recently been shown that $\alpha_2\delta$-1 promotes voltage sensor movement of $Ca_V1.2$ (*Savalli et al., 2016*), thus hyperpolarizing channel activation (*Savalli et al., 2016*). Although the interaction of $\alpha_2\delta$ subunits with $Ca_V2$ channels may differ, as there is not such a clear shift in the voltage-dependence of activation (*Kadurin et al., 2012*; *Canti et al., 2005*), it is tempting to speculate that the association of the channels with uncleaved pro-$\alpha_2\delta$ might interfere with voltage sensor movement.

We have found previously that the MIDAS motif in the VWA domains of $\alpha_2\delta$-1 and $\alpha_2\delta$-2 subunits is key to increasing calcium channel function (*Hoppa et al., 2012*; *Cassidy et al., 2014*; *Canti et al., 2005*). The recent structure of $Ca_V1.1$ indicates that the VWA domain MIDAS motif of $\alpha_2\delta$-1 closely associates with the loop between S1 and S2 in Domain I of the $\alpha1S$ subunit (*Wu et al., 2015*). The VWA domain is present in all $\alpha_2\delta$ subunits, but it only has a perfect MIDAS motif in $\alpha_2\delta$-1 and $\alpha_2\delta$-2 (*Dolphin, 2013*). Thus $\alpha_2\delta$-3 has an incomplete MIDAS motif, which is predicted to have reduced function (*Whittaker and Hynes, 2002*), and indeed $\alpha_2\delta$-3 has less effect on cell surface expression of $Ca_V2.2$ than $\alpha_2\delta$-1 in the present study (compare *Figures 2c* and *4b*).

In agreement with the hypothesis that pro-$\alpha_2\delta$ is an inhibitory gate-keeper for calcium channel function, we observed here that an uncleaved $\alpha_2\delta$-1 construct ($\alpha_2(3C)\delta$-1) had an inhibitory effect on endogenous DRG neuron calcium currents, and on presynaptic $Ca^{2+}$ entry in hippocampal neuronal boutons, which could be reversed by co-expression of the 3C-protease. This is in contrast to a previous study in which overexpression of WT $\alpha_2\delta$ subunits, which will form a mixture of both cleaved and uncleaved $\alpha_2\delta$ (*Figure 6b*), reduced $Ca^{2+}$ entry at synapses (*Hoppa et al., 2012*). Our result

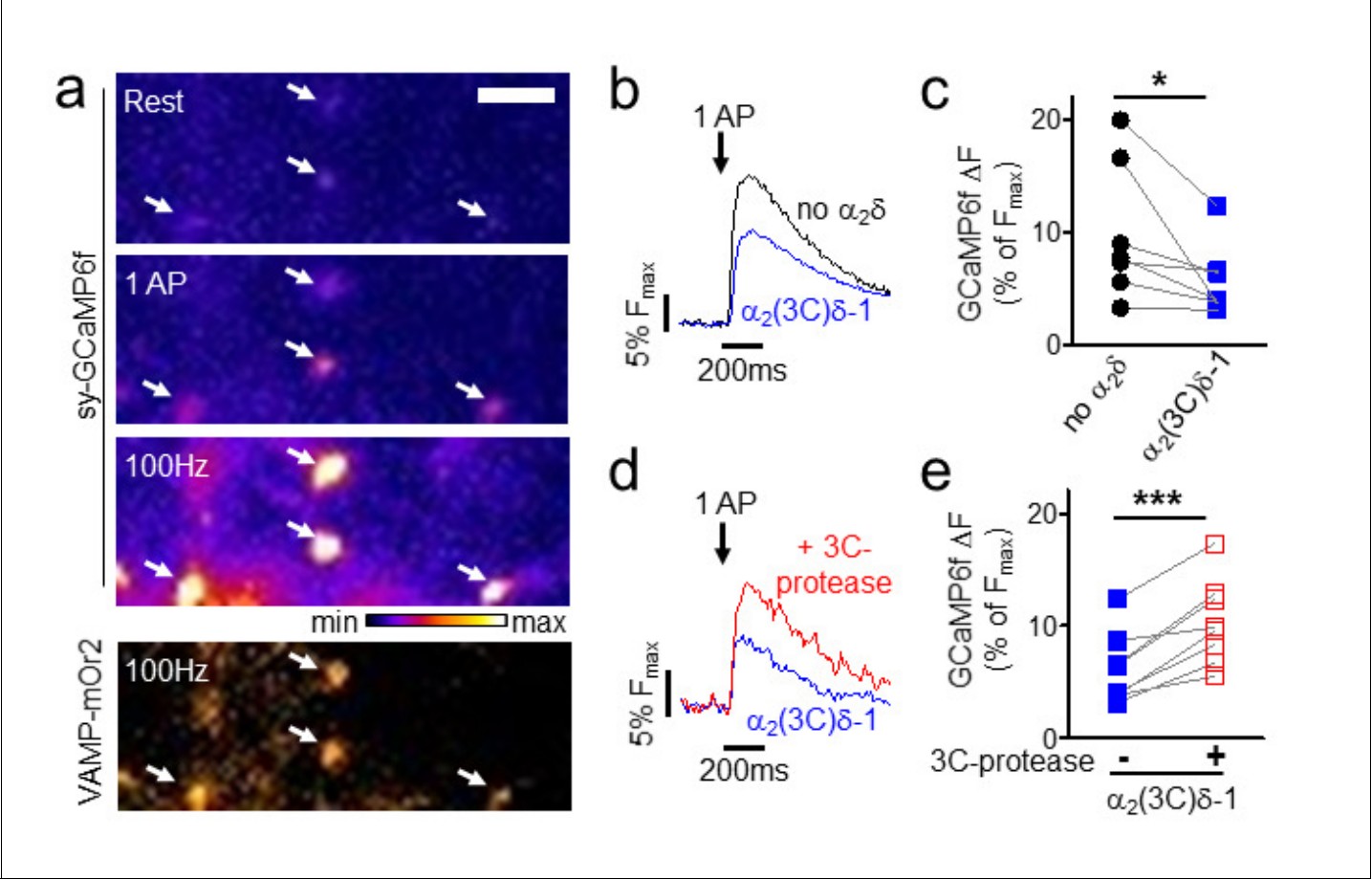

**Figure 8.** Effect of proteolytic cleavage of $\alpha_2(3C)\delta$-1 on $Ca^{2+}$ influx in presynaptic terminals of hippocampal neurons. (**a**) Fluorescence changes in presynaptic terminals of hippocampal neurons expressing sy-GCaMP6f and VAMP-mOr2 in response to electrical stimulation. White arrows point to transfected boutons. Top three panels show sy-GCaMP6f fluorescence: at rest (top), after 1 AP (middle) and after 100 Hz stimulation for 1 s (bottom). The bottom panel shows VAMP-mOr2 fluorescence after 100 Hz stimulation for 1 s. Scale bar 5 μm. The pseudocolour scale is shown below the third panel. (**b**) Mean example traces from the same experiment of sy-GCaMP6f fluorescence changes in response to 5 single APs from individual presynaptic terminals of neurons co-transfected with either empty vector (black trace) or $\alpha_2(3C)\delta$-1 (blue trace). (**c**) Sy-GCaMP6f fluorescence changes (expressed as % of $F_{max}$) in response to 1 AP from boutons co-transfected with either empty vector (black filled circles) or $\alpha_2(3C)\delta$-1 (blue filled squares) (n = 7 independent experiments, *p=0.049, paired t test). (**d**) Mean example traces of sy-GCaMP6f fluorescence changes in response to 5 single APs from presynaptic terminal of neurons co-transfected with either $\alpha_2(3C)\delta$-1 (blue trace) or $\alpha_2(3C)\delta-1$ + 3C-protease (red trace). (**e**) Sy-GCaMP6f fluorescence changes (expressed as % of $F_{max}$) in response to 1 AP from boutons co-transfected with either $\alpha_2(3C)\delta$-1 (blue filled squares) or $\alpha_2(3C)\delta-1$ + 3C-protease (red open squares) (n = 8 independent experiments, ***p=0.0005, paired t test).

indicates that proteolytic processing of $\alpha_2\delta$-1 represents an essential functional checkpoint to allow channel activation by depolarization, and the pro-$\alpha_2\delta$ species inhibits function.

Furthermore, we show here that native pro-$\alpha_2\delta$-1 can be observed in the cell bodies of DRG neurons, but in the axons it is completely processed to $\alpha_2$ and $\delta$; this is despite the fact that it is still present in intracellular trafficking vesicles (*Bauer et al., 2009*). It is of interest in this regard that although $\alpha_2\delta$-1 is elevated in all DRG neurons following nerve injury (*Bauer et al., 2009*; *Luo et al., 2001*), calcium currents from DRGs extracted after nerve injury show a variable decrease in calcium currents (*Hogan et al., 2000*; *McCallum et al., 2006, 2011*). This would agree with the present finding that somatic DRG $\alpha_2\delta$-1 upregulated after nerve injury remains in part uncleaved, and that only when in its mature processed form can it function to increase calcium currents and trafficking of calcium channels into neurites.

The functional importance of proteolytic cleavage of $\alpha_2\delta$ subunits is further emphasised by our finding that the trafficking of $Ca_V2.2$ into the processes of cultured hippocampal neurons is

completely prevented by the uncleaved $\alpha_2$(3C)$\delta$-1, and this is reversed by its intracellular proteolytic cleavage with the 3C protease. Thus, the proteolytic processing of $\alpha_2\delta$ represents an essential checkpoint for neuronal trafficking of calcium channels, to ensure that only mature channel complexes capable of voltage-dependent activation reach specific plasma membrane domains such as presynaptic terminals (*Figure 7—figure supplement 1*). We have shown previously that the small GTPase, rab11, is involved in $\alpha_2\delta$-mediated trafficking of calcium channels (*Tran-Van-Minh and Dolphin, 2010*), and rab11, among many other proteins, is important for vesicular cargo transport into neurites (*Villarroel-Campos et al., 2016*). We have also identified the importance of adaptor protein-1 in Ca$_V$2.2 trafficking (*Macabuag and Dolphin, 2015*). It will be of great interest in future studies to determine the additional mechanisms present in neurons, in contrast to non-neuronal cell lines, restricting the cargo transport into neurites to activatable calcium channel complexes in which $\alpha_2\delta$ subunits are proteolytically cleaved.

# Materials and methods

## Molecular biology

The cDNAs used were: rat $\alpha_2\delta$-1 (M86621) and mouse $\alpha_2\delta$-3 (AJ010949), rabbit Ca$_V$2.2 (D14157 without 3' UTR), and rat $\beta$1b (*Tomlinson et al., 1993*). In some experiments Ca$_V$2.2 was used with an N-terminal GFP (*Macabuag and Dolphin, 2015*), or containing an extracellular BBS tag or HA tag (*Cassidy et al., 2014*). $\alpha_2\delta$-1-HA (HA-tag sequence YPYDVPDYA inserted between Asn-549 and Asp-550) (*Kadurin et al., 2012*) and $\alpha_2\delta$-3-HA (HA between Lys-595 and Arg-596) were used in all imaging and other experiments except when the subunits were co-expressed with Ca$_V$2.2-HA in hippocampal neurons, in which case untagged $\alpha_2\delta$-1 or $\alpha_2\delta$-3 were used. Both $\alpha_2\delta$-HA constructs showed normal function in terms of enhancing Ca$_V$ currents ([*Kadurin et al., 2012*] and data not shown). Proteolytic cleavage site mutations were made as indicated (*1b* and *4c*). GFP-$\beta$1b was used in some electrophysiological experiments (*Waithe et al., 2011*). Human TASK3 (*KCNK9*) cDNA (NM_001282534) was obtained from Prof. A Mathie. The cDNAs were in the pMT2 vector for expression in tsA-201 cells, in pcDNA3 for expression in N2A cells and pcDNA3 or pCAGGS for expression in DRG or hippocampal neurons, respectively. In our hands expression of the large constructs encoding Ca$_V$2.2 and $\alpha_2\delta$ in pcDNA3 was very poor in hippocampal neurons (data not shown), whereas expression from the pCAGGS vector, containing chicken $\beta$-actin promoter, was strong and sustained. The cDNAs encoding Human Rhinovirus (HRV)$-$3C protease or mutated 3C (in which the active site Cys-147 was mutated to Val) were first subcloned into the pHLSec vector (*Yurtsever et al., 2012*; *Aricescu et al., 2006*), using Age I and Kpn I sites in frame with an N-terminal signal sequence, to include the signal sequence from pHLSec, and then into pMT2 and pCAGGS vectors for expression in cell lines or hippocampal neurons. VAMP-mOr2 was generated by replacing mCherry from pCAGGs-VAMPmCherry (gift from Dr. TA Ryan) with mOrange2. Sy-GCaMP6f was made by replacing GCaMP3 in pCMV-SyGCaMP3 (gift from Dr. TA Ryan) by GCaMP6f (*Chen et al., 2013*). The cDNA for mut2-GFP (*Cormack et al., 1996*) in pMT2 was used as a negative control in co-immunoprecipitation (co-IP) experiments. Site-directed mutagenesis was carried out using standard procedures, and all subcloning and mutations confirmed by sequencing.

## Antibodies and other materials

Ca channel antibodies (Abs) used were: $\alpha_2\delta$-1 Ab (mouse monoclonal against $\alpha_2$-1 moeity, Sigma-Aldrich, epitope identified in [*Cassidy et al., 2014*]), $\alpha_2$-3 (71–90) Ab (rabbit; polyclonal) (*Davies et al., 2010*), $\delta$-3 (1035–1049) Ab (*Davies et al., 2010*), anti-Ca$_V$2.2 II-III loop Ab (rabbit polyclonal) (*Raghib et al., 2001*). Other Abs used were anti-HA (rat monoclonal, Roche), anti-HA (rabbit polyclonal, Sigma), anti-GAPDH Ab (mouse monoclonal, Ambion), anti-flotillin-1 (mouse monoclonal, BD Biosciences), and GFP Ab (Living Colors, rabbit polyclonal; BD Biosciences). For immunocytochemistry, secondary Abs (1:500) used were anti-rabbit-Alexa Fluor 594, anti-rat-Alexa Fluor 594, anti-mouse-Alexa Fluor 647 (Life Technologies) or fluorescein isothiocyanate (FITC)-anti-rat (Sigma-Aldrich). The following secondary Abs were used for western blot: goat anti-rabbit, goat anti-rat and goat anti-mouse Abs coupled to horseradish peroxidase (HRP) (Biorad, Hemel Hempstead, UK). The signal was obtained by HRP reaction with fluorescent product (ECL 2; Thermo Scientific) and membranes were scanned on a Typhoon 9410 phosphorimager (GE Healthcare).

Lyophilized active thrombin was obtained from Sigma, suspended in filter-sterilised PBS (Sigma; pH7.4) to 1000 U/ml and frozen in aliquots until use.

## Cell culture, transfection and enzymatic treatment

Cell lines were plated onto cell culture flasks, coverslips or glass-bottomed dishes (MatTek Corporation, Ashland, MA), coated with poly-L-lysine, and cultured in a 5% $CO_2$ incubator at 37°C. The human embryonic kidney tsA-201 cells (European Collection of Authenticated Cell Cultures, # 96121229), tested to be mycoplasma-free, were cultured in Dulbecco's modified Eagle's medium (DMEM) supplemented with 10% foetal bovine serum (FBS), 1 unit/ml penicillin, 1 µg/ml streptomycin and 1% GlutaMAX (Life Technologies, Waltham, MA). tsA-201 cells were transfected using Fugene6 (Promega, Fitchburg, WI), according to the manufacturer's protocol. The enzymatic treatments with 30 U/ml Thrombin protease (Sigma) diluted in DMEM without supplements were done for indicated times in a 5% $CO_2$ incubator at 37°C. Mouse neuroblastoma N2A cells (American Tissue culture collection, # CCL-131) were obtained from Professor Roger Morris, Kings College London (*Parkyn et al., 2008*), and were tested to be mycoplasma-free. They were cultured in 50% DMEM and 50% OPTI-MEM supplemented with 5% FBS, 1 unit/ml penicillin, 1 µg/ml streptomycin, and 1% GlutaMAX. N2A cells were transfected using PolyJet (SignaGen Laboratories, Gaithersburg, MD), according to the manufacturer's protocol.

## Neuronal culture and transfection

DRG neurons were isolated from P10 male Sprague Dawley rats and transfected essentially as described recently (*D'Arco et al., 2015*) with an Amaxa Nucleofector (Lonza, Basel, Switzerland) according to the manufacturer's protocol. Transfected neurons were plated onto coverslips coated with poly-L-lysine, and cultured in DMEM/F12 supplemented with 10% FBS, 1 unit/ml penicillin, 1 µg/ml streptomycin, 1% GlutaMAX and 100 ng/ml NGF in a 5% $CO_2$ incubator at 37°C. Hippocampal neurons were obtained from P0 rat pups as previously described (*Morales et al., 2000*). Approximately $75 \times 10^3$ cells in 100 µl of plating solution (Neurobasal medium supplemented with B27 (Life Technologies; 2%), HEPES (10 mM), horse serum (5%), glutamine (0.5 mM), and 1 unit/ml penicillin, 1 µg/ml streptomycin) were seeded onto sterile poly-lysine-coated glass coverslips. After 2 hr, the plating solution was replaced with 1 ml of growth medium (serum-free Neurobasal medium supplemented with B27 (Life Technologies; 4%), 2-mercaptoethanol (25 µM), glutamine (0.5 mM), and 1 unit/ml penicillin, 1 µg/ml streptomycin), half of which was replaced every 3–4 days. At 7 days in vitro (DIV) and 2 hr before transfection, half of the medium was removed, and kept as 'conditioned' medium and 500 µl of fresh medium was added. The hippocampal cell cultures were then transfected with Lipofectamine 2000 according to the manufacturer's protocol using 2 µg of cDNA mix per well (Invitrogen). After 2 hr, the transfection mixes were replaced with growth medium consisting of 50% 'conditioned' and 50% fresh medium. The cultures were used for immunostaining experiments at 14 DIV, or for live imaging as described below.

## Cell surface biotinylation, Cell lysis, deglycosylation and immunoblotting

The procedures were modified from those described in more detail previously (*Davies et al., 2010*; *Page et al., 2004*). Briefly, 72 hr after transfection, tsA-201 cells were incubated for 30 min at room temperature with 0.5 mg/ml Premium Grade EZ-link Sulfo-NHS-LC-Biotin (Thermo Scientific) in PBS and the reaction was quenched with 200 mM glycine. The cells were incubated for 45 min on ice in PBS, pH 7.4 at 4°C containing 1% Igepal; 0.1% SDS and protease inhibitors (PI, cOmplete, Roche; used according to manufacturer's instructions), to allow cell lysis. The WCL was then centrifuged at 18,000 ×g for 20 min at 4°C and the pellet discarded. The supernatant was assayed for total protein (Bradford assay, Biorad). Immunoblot analysis was performed essentially as described previously (*Kadurin et al., 2012*). Cleared WCL corresponding to 20–40 µg total protein was diluted with Laemmli sample buffer (*Davies et al., 2010*) supplemented with 100 mM dithiothreitol (DTT), incubated at 60°C for 10 min and resolved by SDS-polyacrylamide gel electrophoresis (PAGE) on 3%–8% Tris-Acetate or 4–12% Bis-Tris gels (Invitrogen) and transferred to polyvinylidene fluoride (PVDF) membrane (Biorad) by western blotting. When required the membrane was cut and incubated with different antibodies. Biotinylated lysates (adjusted to between 0.5 and 1 mg/ml total protein

concentration) were applied to 40 µl prewashed streptavidin-agarose beads (Thermo Scientific) and rotated overnight at 4°C. The beads were then washed 3 times with PBS containing 0.1% Igepal and, when required, the streptavidin beads were re-suspended in PNGase-F buffer (PBS, pH 7.4, supplemented with 75 mM β-mercaptoethanol, 1% Igepal, 0.1% SDS, and PI) and deglycosylated for 3 hr at 37°C with 1 unit of PNGase-F (Roche Applied Science) added per 10 µl volume. The samples were then resuspended in an equal volume of 2 × Laemmli buffer with 100 mM DTT, followed by 10 min incubation at 60°C. The eluted protein was analysed by immunoblotting, as described above. GAPDH is ~39 kDa and does not resolve well in 3–8% Tris-Acetate gels. Therefore, in cases when 3–8% gels were used to resolve high MW proteins, which required GAPDH as negative control for biotinylated fractions, the same samples were re-loaded on a 4–12% Bis-Tris gel to analyse separately.

## Co–immunoprecipitation

The protocol was adapted from a procedure described previously (*Gurnett et al., 1997*). Briefly, a tsA-201 cell pellet derived from one confluent 75-cm$^2$ flask was resuspended in a co-IP buffer (20 mM HEPES (pH 7.4), 300 mM NaCl, 1% Digitonin and PI), sonicated for 8 s at 20 kHz and rotated for 1 hr at 4°C. The samples were then diluted with an equal volume of 20 mM HEPES (pH 7.4), 300 mM NaCl with PI (to 0.5% final concentration of Digitonin), mixed by pipetting and centrifuged at 18,000 ×g for 20 min. The supernatants were collected and assayed for total protein (Bradford assay; Bio-rad). 1 mg of total protein was adjusted to 2 mg/ml with co-IP buffer and incubated overnight at 4°C with anti-GFP polyclonal antibody (1:200; BD Biosciences). 30 µl A/G PLUS Agarose slurry (Santa Cruz) was added to each tube and further rotated for 2 hr at 4°C. The beads were then pelleted by 500 ×g centrifugation at 4°C and washed three times with co-IP buffer containing 0.2% Digitonin. The beads were then resuspended in an equal volume of PNGase-F buffer and proteins were deglycosylated with PNGase-F as described above. Aliquots of the initial WCL prior to co-IP were also deglycosylated in parallel; 2 × Laemmli buffer with 100 mM DTT was added and samples were analysed by SDS-PAGE and western blotting.

## Preparation of triton X-100-insoluble membrane fractions (DRMs)

The protocol was similar to that described previously (*Davies et al., 2010*; *Kadurin et al., 2012*). All steps were performed on ice. Confluent tsA-201 cells from two 175-cm (*Kurshan et al., 2009*) flasks were taken up in Mes-buffered saline (MBS, 25 mm Mes, pH 6.5, 150 mm NaCl, and PI) containing 1% (v/v) Triton X-100 (TX-100) (Thermo Scientific), and left on ice for 1 hr. An equal volume of 90% (w/v) sucrose in MBS was then added to a final concentration of 45% sucrose. The sample was transferred to a 13 ml ultracentrifuge tube and overlaid with 10 ml of discontinuous sucrose gradient, consisting of 35% (w/v) sucrose in MBS (5 ml) and 5% (w/v) sucrose in MBS (5 ml). The sucrose gradients were ultra-centrifuged at 140,000 ×$g_{avg}$ (Beckman SW40 rotor) for 18 hr at 4°C. 1 ml fractions were subsequently harvested from the top to the bottom of the tube and aliquots of 10 µl from each fraction were analysed by SDS-PAGE and western blotting to obtain DRM profiles. When necessary, DRMs (combined peak fractions identified by the presence of flotillin-1) from the gradient were washed free of sucrose by dilution into 25 volumes of cold PBS (pH 7.4) and pelleted by ultra-centrifugation at 150,000 ×$g$ (Beckman Ti 70 rotor) for 1 hr at 4°C. TX-100-insoluble protein was resuspended in appropriate buffers as described for $^3$H-gabapentin binding or for deglycosylation as described above.

## Immunocytochemistry, imaging and analysis

The procedure in tsA-201 and N2A cells was performed essentially as described previously with minor modifications (*Davies et al., 2010*; *Kadurin et al., 2012*). Briefly, 48 hr post-transfection the cells were fixed with 4% paraformaldehyde (PFA) in PBS, pH7.4 at 20°C for 5 min, and then incubated for PBS for 15 min, which contained 0.1% TX-100 if permeabilization was applied. Blocking was performed for 1 hr at 20°C in PBS containing 20% goat serum and 5% bovine serum albumen (BSA). The indicated primary antibodies were then applied (diluted in PBS with10% goat serum and 2.5% BSA) overnight at 4°C or for 1 hr at 20°C. In live-labelling experiments, cells were washed with Krebs Ringer HEPES (KRH) buffer, labelled with α-bungarotoxin (BTX)-AF 488 (Invitrogen; 1:100 in KRH buffer) at 17°C for 30 min, then washed with KRH and fixed as described above. The indicated secondary antibodies were applied (1:500 dilution in PBS, containing 2.5% BSA and 10% goat serum)

at 20°C for 1 hr. Cell nuclei were stained with 0.5 μM 4′,6′-diamidino-2-phenylindole (DAPI) in PBS for 5 min. The coverslips were mounted onto glass slides using VECTASHIELD mounting medium (Vector Laboratories, Peterborough, UK). Cultures of transfected hippocampal neurons were fixed after 14 DIV in PBS containing 4% PFA/4% sucrose for 5 min at 20°C, and then the procedure was as described above. In some cases, where stated, an antigen retrieval step was performed between the fixation and blocking steps: the cells were incubated for 10 min at 95°C in 10 mM citrate buffer (pH 6) containing 0.05% Tween 20.

Imaging was performed on Zeiss LSM 780 confocal microscope as described in more detail elsewhere (*Davies et al., 2010*; *Kadurin et al., 2012*). Images were obtained at fixed microscope settings for all experimental conditions of each experiment. Images of N2A and tsA-201 cells were obtained using a 63 × objective at a resolution of 1024 × 1024 pixels and an optical section of 0.8–1 μm. After choosing a region of interest containing transfected cells, the 3 × 3 tile function of the microscope allowed imaging of a larger area selected without bias. Every cell identified as transfected was included in the measurements, to ensure lack of bias.

Images of tsA-201 and N2A cells were analyzed using imageJ (*imagej.net*) using a modification of the procedure described previously (*Davies et al., 2010*; *Kadurin et al., 2012*). Surface labelling in non-permeabilized or total staining in permeabilized cell bodies was measured using the freehand line tool and manually tracing the surface of the cell or drawing around the cell (omitting the nucleus) respectively. The value of the mean pixel intensity in different channels was measured separately and background was subtracted by measuring the intensity of an imaged area without transfected cells. All data were then normalized to the appropriate positive control for each experiment before combining experiments.

Hippocampal neurons were imaged using a 20 × objective with a 5 μm optical section. Large tiles were manually selected following all processes expressing mCherry. The fluorescence intensity along neuronal projections of hippocampal neurons was assessed in FIJI as follows: a circle of 100 μm diameter was drawn around each neuronal cell body. A free-hand line (2 μm thick; ~ 30 μm long) was drawn along the neurite extending beyond the circle and the mean grey intensity of all the pixels within the line was measured in both channels corresponding to the fluorescence of HA or $\alpha_2\delta$-1 immunostaining and mCherry.

## Analysis of $\alpha_2\delta$-1 in dorsal root ganglion (DRG) neurons and axons

Immunoblotting of DRG tissue and associated nerves was performed as described previously (*Bauer et al., 2009*). Tissue was taken from rats 4d following a spinal nerve ligation procedure, performed in the course of a previous study (*Bauer et al., 2009*), in order to increase the amount of $\alpha_2\delta$-1 protein in the harvested tissue. Tissue from DRGs, sections of spinal nerve and dorsal roots were pooled from 4 rats, and stored at −80°C until use in this study. For deglycosylation, protein samples were diluted in PBS + 1% Igepal + 1% β-mercaptoethanol and treated with 1 unit of PNGase F overnight at 37°C.

## $^3$H gabapentin binding assay

Binding of $^3$H-gabapentin to DRM preparations was carried out, essentially as previously described (*Lana et al., 2014*), in a final volume of 250 μl at room temperature for 45 min. DRM fractions (4 μg of protein per tube) were incubated with various concentrations of [$^3$H]-gabapentin (specific activity 36 Ci/mmol, American Radiolabeled Chemicals, St. Louis, MO, USA) in 10 mM HEPES/KOH pH 7.4, then rapidly filtered through GF/B filters, pre-soaked with 0.3% polyethyleneimine. Filters were washed three times with 3 ml ice-cold 50 mM Tris/HCl, pH 7.4 and counted on a scintillation counter. Concentrations of [$^3$H]-gabapentin greater than 50 nM were achieved by adding non-radioactive gabapentin and correcting the specific binding by the dilution factor (*Canti et al., 2005*; *Lana et al., 2014*). Non-specific binding was determined in the presence of 20 μM non-radioactive gabapentin. Data points were determined in triplicate for each experiment, and data for each experiment were analysed by fitting specific binding to the Hill equation (*Lana et al., 2014*).

## Electrophysiology

Calcium channel currents in transfected tsA-201 cells were investigated by whole cell patch clamp recording, essentially as described previously (*Berrow et al., 1997*). The patch pipette solution

contained in mM: Cs-aspartate, 140; EGTA, 5; MgCl$_2$, 2; CaCl$_2$, 0.1; K$_2$ATP, 2; Hepes, 10; pH 7.2, 310 mOsm with sucrose. The external solution for recording Ba$^{2+}$ currents contained in mM: tetrae-thylammonium (TEA) Br, 160; KCl, 3; NaHCO$_3$, 1.0; MgCl$_2$, 1.0; Hepes, 10; glucose, 4; BaCl$_2$, 1,or 2 as indicated, pH 7.4, 320 mosM with sucrose. Unless otherwise stated, 1 mM extracellular Ba$^{2+}$ was the charge carrier. Pipettes of resistance 2–4 MΩ were used. An Axopatch 1D or Axon 200B amplifier was used, and whole cell voltage-clamp recordings were sampled at 10 kHz frequency, filtered at 2 kHz and digitized at 1 kHz. 70–80% series resistance compensation was applied and all recorded currents were leak subtracted using P/8 protocol. For DRG neurons, whole cell voltage clamp experiments were performed in small (<19 pF) and medium (20–38 pF) neurons. Membrane potential was held at −80 mV for experiments in tsA-201 cells and −90 mV for DRG experiments. Cells were accepted where the access resistance was less than 5 MΩ, the inward current was > −3 pA/pF at +10 mV, a complete *IV* relationship was obtained and there was no evidence of poor voltage clamp. Analysis was performed using Pclamp 9 (Molecular Devices) and Origin 7 (Microcal Origin, Northampton, MA). *IV* relationships were fit by a modified Boltzmann equation as follows: $I=G_{max}*(V-V_{rev})/(1+exp(-(V-V_{50, act})/k))$ where *I* is the current density (in pA/pF), $G_{max}$ is the maximum conductance (in nS/pF), $V_{rev}$ is the apparent reversal potential, $V_{50, act}$ is the midpoint voltage for current activation, and *k* is the slope factor. Recordings of resting membrane potential were performed as previously described (*Margas et al., 2016*).

## Live cell imaging

Hippocampal neurons were transfected with VAMP-mOr2 and sy-GCaMP6f, together with the other cDNAs used at 7 DIV. Neurons were imaged after 14–21 DIV. Coverslips were mounted in a laminar-flow perfusion and stimulation chamber (Warner Instruments) on the stage of an epifluorescence microscope (Axiovert 200 M, Zeiss). White and 470 nm LEDs served as light sources (Cairn Research, UK). Fluorescence excitation and collection was performed through a 40 × 1.3 NA Fluar Zeiss objective using 450/50 nm excitation and 510/50 nm emission and 480 nm dichroic filters, and a 545/25 nm excitation and 605/70 nm emission and 565 nm dichroic filters (for mOrange2). Live cell images were acquired as previously described with minor modifications (*Margas et al., 2016*; *Ferron et al., 2014*) with an Andor iXon+ (model DU-897U-CS0-BV) back-illuminated EMCCD camera. Fluorescence was collected at 100 Hz over a 512 × 266 pixel area (7 ms integration time). Cells were perfused (0.5 ml min$^{-1}$) in a saline solution at 22°C containing (in mM) 119 NaCl, 2.5 KCl, 2 CaCl$_2$, 2 MgCl$_2$, 25 HEPES (buffered to pH 7.4), 30 glucose, 10 μM 6-cyano-7-nitroquinoxaline-2,3-dione (CNQX) and 50 μM D,L-2-amino-5-phosphonovaleric acid (AP5). Neurons were stimulated by passing 1 ms current pulses through the field stimulation chamber via platinum electrodes. Neurons expressing syn-GCaMP6f were identified by stimulating the preparation at 33 Hz for 180 ms every 4 s. Subsequently, single stimulations of 1 ms (mimicking a single AP) were repeated 5 times at 45 s intervals. Functional synaptic boutons were identified by the increase of fluorescence of VAMP-mOr2 in response to a 100 Hz stimulation for 1 s. Changes in GCaMP6f fluorescence were normalized to the maximum fluorescence (F$_{max}$) measured in the presence of 5 μM ionomycin and 2 mM Ca$^{2+}$ in the perfusion solution. Analysis was performed with ImageJ (http://rsb.info.nih.gov/ij), using a custom-written plugin (http://rsb.info.nih.gov/ij/plugins/time-series.html). Regions of interest (ROI) of 2 μm diameter were selected according to their responsiveness to a 100 Hz stimulation for 1 s (on average, 20 to 100 ROIs were analyzed per field of view). Peak fluorescence in response to mean of 5 single AP stimuli was determined by averaging 5–10 points of the plateau phase and subtracting the average of 10 points of the baseline before stimulation.

## Data analysis

Data are given as mean ± SEM, or scatter plots, or box (25–75%) and whisker (10–90%) plots with mean and median (dashed and solid lines). Statistical comparisons were performed using either Student's t test, paired t test, 1-sample t test, ANOVA with appropriate post-hoc test, or Krushkal-Wallis test with appropriate post-hoc test, as stated, using Graphpad Prism 5. Details of tests are given in Supplementary statistics file.

## Acknowledgements

This work was supported by a Wellcome Trust Investigator award to ACD (098360/Z/12/Z) and Medical Research Council (UK) (grants G0901758 and G0801756). We thank Kanchan Chaggar for tissue culture. We thank Prof. AH Dickenson and Dr. Wahida Rahman for the collaboration *Bauer et al., 2009* in which tissue for *Figure 6a* was collected. We thank Dr. Matthew Gold for HRV-3C protease cDNA, Prof. Alistair Mathie for TASK3 cDNA and Prof Tim Ryan for VAMP-mCherry and Synaptophysin-GCaMP3 cDNAs.

## Additional information

### Funding

| Funder | Grant reference number | Author |
| --- | --- | --- |
| Wellcome Trust | 098360/Z/12/Z | Ivan Kadurin<br>Laurent Ferron<br>Simon W Rothwell<br>Wojciech Margas<br>Manuela Nieto-Rostro |
| Medical Research Council | G0901758 | Laurent Ferron<br>Leon R Douglas<br>Claudia S Bauer |
| Medical Research Council | G0801756 | Laurent Ferron<br>Leon R Douglas<br>Claudia S Bauer |
| Medical Research Council | MR/J013285/1 | Laurent Ferron<br>Leon R Douglas<br>Claudia S Bauer |

The funders had no role in study design, data collection and interpretation, or the decision to submit the work for publication.

### Author contributions

IK, LF, Conception and design, Acquisition of data, Analysis and interpretation of data, Drafting or revising the article; SWR, JOM, LRD, BL, WM, Acquisition of data, Analysis and interpretation of data; CSB, Acquisition of data, Analysis and interpretation of data, Drafting or revising the article; OA, Acquisition of data, Contributed unpublished essential data or reagents; MN-R, Conception and design, Acquisition of data, Analysis and interpretation of data; WSP, Conception and design, Contributed unpublished essential data or reagents; ACD, Conception and design, Analysis and interpretation of data, Drafting or revising the article

### Author ORCIDs

Annette C Dolphin, ![ORCID] http://orcid.org/0000-0003-4626-4856

## Additional files

### Supplementary files

• Supplementary file 1. Summary of statistics shown for the quantitative data in the Figures. The statistical data were obtained using Graphpad Prism 5.

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
