## [Decision Letter]

Thank you for submitting your article "Proteolytic maturation of α_2_δ represents a checkpoint for activation and neuronal trafficking of latent calcium channels" for consideration by *eLife*. Your article has been reviewed by 2 peer reviewers, including Johannes Hell (Reviewer #3), and the evaluation has been overseen by Mary Kennedy as Reviewing Editor and Richard Aldrich as the Senior Editor.

The reviewers have discussed the reviews with one another and the Reviewing Editor has drafted this decision to help you prepare a revised submission.

Summary:

In this manuscript, Kadurin et al. probe the significance of proteolytic processing of the auxiliary α_2_δ subunit of voltage-gated calcium channel Ca_v_2.2 for its trafficking to the cell surface and to the presynaptic compartment and for its role in regulation of channel activity. They find that a hexa-valine mutation of the putative proteolytic cleavage site in α_2_δ prevents proteolytic processing but allows cell surface expression of the α_2_δ subunit and normal binding to the Ca_v_2.2 channel. Surprisingly, proteolytic processing of α_2_δ is required for voltage-dependent activation, indicating a functional role in ion channel regulation for this posttranslational processing reaction. Addition of an exogenous proteolytic cleavage site allows activation of α_2_δ function by exogenous protease. Proteolytic processing can occur after insertion into the plasma membrane and can rescue inactive Ca_v_2.2 channels there. In cultured DRG neurons, α_2_δ subunits are processed in intracellular compartments, and their proteolytic processing controls trafficking into neurites and synaptic boutons, cell surface expression, and voltage-dependent activation of calcium channel function in nerve terminals. Thus, the authors describe three novel physiological aspects of the function of the α_2_δ subunit of voltage-gated Ca channels in the context of the N-type channel Ca_v_2.2. Firstly, α_2_δ strongly augments open probability of Ca_v_2.2 independent of its role in promoting surface trafficking. Secondly, cleavage of the uncleaved pro-α_2_ is required for this effect, as shown by extremely thorough experiments including acute application of thrombin to a modified α_2_δ in which the endogenous cleavage site has been replaced by a thrombin cleavage site. Thirdly, uncleaved pro-α_2_ inhibits secretory trafficking of Ca_v_2.2 in neurons.

This is a very important manuscript because it shows how an auxiliary subunit of a presynaptic calcium channel can control channel localization and function through posttranslational processing. Work from the Dolphin lab and others has shown how the α_2_δ subunit enhances trafficking in transfected cells and neurons. This manuscript adds a major new dimension by showing that proteolytic processing of this subunit is required for both trafficking to nerve terminals and for activation of the channel at the cell surface. The work is comprehensive and exceptionally well done. The findings are of broad interest, not least because upregulation of Ca_v_2.2, and especially its association with α_2_δ, is a critical aspect of neuropathic pain.

Essential revisions:

1) The site of proteolytic processing of α_2_δ was shown in two manuscripts in 1990 (JBC 265:14738) and 1991 (JBC 266:3287) to be at Ala934, with the δ subunit beginning with the N-terminal sequence ADMEDDD. Both papers should be cited.

2) Surprisingly, it seems that the V6 mutation and other mutations used here do not overlap the reported site of cleavage for α_2_δ-1. Has the site of cleavage been verified for the amino acid sequences shown here? Reviewer #3 commented that the original work in rabbit muscle identified the sequence ADMEDDD as the N-terminal sequence of the δ subunit, which is cleaved off the α_2_δ precursor. The preceding sequence in the rabbit (and rat) gene is PRLLEA. Accordingly, the cleavage site consists in rabbit of PRLLEA//ADMEDDD and in rat of PRLLEA//VDMEEDD. It seems to me that this is reasonably good conservation – there are only two conservative differences (Valine instead of Alanine and then in position 5 after cleavage Aspartate instead of Glutamate. The rat sequence is (PRLLEA//VDMEEDD) and this is 100% identical in many other mammals including donkey and camelids. The authors might want to discuss this conservation issue and verify the site of cleavage.

3) It is not immediately clear to the general reader why in the HEK-derived tsA cells non-cleavable α_2_ would (modestly) promote Ca_v_2.2 surface expression comparable to wild type α_2_δ when in neurons it clearly impairs trafficking. Perhaps the reason is similar to what one sees for the L-type channel Ca_v_1.2 – most of the channel is stuck inside the cell and doesn't seem to readily traffic to the surface with the exception of a small fraction of the channel population that escapes ER and that escape could be promoted by α_2_δ in a manner that doesn't apply to endogenous systems? It might be difficult to put a finger on the exact reason but perhaps some more explicit discussion would be helpful. This issue doesn't bear on the main findings, which are completely supported by the current work.

---

## [Author Response]

We thank the Editor and reviewers for their comments. One point we would like to make clear is that our initial studies using the hexavaline mutations in α_2_δ-1 and α_2_δ-3 mainly served as proof of concept that we could disrupt their cleavage sites. These constructs were not used in cell surface expression studies. In all subsequent studies we used α_2_δ subunits with either of the artificial cleavage sites (HRV-3C or thrombin).

*[…] Essential revisions: 1) The site of proteolytic processing of α2δ was shown in two manuscripts in 1990 (JBC 265:14738) and 1991 (JBC 266:3287) to be at Ala934, with the δ subunit beginning with the N-terminal sequence ADMEDDD. Both papers should be cited.*

We apologise for our oversight in omitting to cite De Jongh et al. (1990). This is now included together with Jay et al. (1991).

*2) Surprisingly, it seems that the V6 mutation and other mutations used here do not overlap the reported site of cleavage for α2δ-1. Has the site of cleavage been verified for the amino acid sequences shown here? Reviewer #3 commented that the original work in rabbit muscle identified the sequence ADMEDDD as the N-terminal sequence of the δ subunit, which is cleaved off the α2δ precursor. The preceding sequence in the rabbit (and rat) gene is PRLLEA. Accordingly, the cleavage site consists in rabbit of PRLLEA//ADMEDDD and in rat of PRLLEA//VDMEEDD. It seems to me that this is reasonably good conservation – there are only two conservative differences (Valine instead of Alanine and then in position 5 after cleavage Aspartate instead of Glutamate. The rat sequence is (PRLLEA//VDMEEDD) and this is 100% identical in many other mammals including donkey and camelids. The authors might want to discuss this conservation issue and verify the site of cleavage.*

The polyvaline and other mutations used *do* overlap the predicted cleavage site in α_2_δ-1 (see alignment in Figure 1), which as one reviewer mentioned is quite well conserved between rabbit and rat. We now discuss that the cleavage site in rabbit α_2_δ-1 was identified (/ in the sequence below) by N-terminal sequencing of the δ-1 peptides, which identified ADM at the start of all three δ-1 peptides. The sequence in rat α_2_δ-1 used here is aligned below to some other mammalian species.

NP_001075745 *Oryctolagus cuniculus* TFPRLLEA/ADMEDDDFTASMSK

NP_037051.2 *Rattus norvegicus* TFPRLLEA VEMEEDDFTASMSK

XP_010976290 *Camelus dromedaries* TFPRLLEA VEMEEDDFTASLSK

XP_014710873 *Equus asinus* TFPRLLEA VEMEEDDFTASLSK

From the good sequence alignment, identical before the cleavage site, we predicted the cleavage site in rat α_2_δ-1 to be between A and V. In order to clarify this point, further details, including the sequences from camel and donkey are now given in the Figure 1—figure supplement 1. We thank the reviewer for this insight, and have commented on the sequence conservation in other mammals.

*3) It is not immediately clear to the general reader why in the HEK-derived tsA cells non-cleavable α2 would (modestly) promote Cav2.2 surface expression comparable to wild type α2δ when in neurons it clearly impairs trafficking. Perhaps the reason is similar to what one sees for the L-type channel Cav1.2 – most of the channel is stuck inside the cell and doesn't seem to readily traffic to the surface with the exception of a small fraction of the channel population that escapes ER and that escape could be promoted by α2δ in a manner that doesn't apply to endogenous systems? It might be difficult to put a finger on the exact reason but perhaps some more explicit discussion would be helpful. This issue doesn't bear on the main findings, which are completely supported by the current work.*

We were also surprised to find that non-cleavable α_2_δ-1 promotes trafficking of Ca_V_2.2 in both N2A cells, and to a slightly smaller extent in tsA-201 cells. We suggest this is because there are fewer check points in non-neuronal cells for channel trafficking, which do not apply in neurons. Initially we hypothesised that a difference in membrane potential might be the key feature, discriminating between neurons and non-neuronal cells, but this did not prove to be the case (Figure 2). We have now discussed this point further in the Discussion, with respect to the presence of additional checkpoints in neurons for cargo selection into specific trafficking pathways.